# Core Advantage Decomposition for Policy Gradients in Multi-Agent Reinforcement Learning

## Abstract

This work focuses on the credit assignment problem in cooperative multi-agent reinforcement learning (MARL). Sharing the global advantage among agents often leads to insufficient policy optimization, as it fails to capture the coalitional contributions of different agents. Existing methods mainly assign credits based on individual counterfactual contributions, while overlooking the influence of coalitional interactions. In this work, we revisit the policy update process from a coalitional perspective and propose an advantage decomposition method guided by the cooperative game-theoretic core solution. By evaluating marginal contributions of all possible coalitions, our method ensures that strategically valuable coalitions receive stronger incentives during policy gradient updates. To reduce computational overhead, we employ random coalition sampling to approximate the core solution efficiently. Experiments on matrix games, differential games, and multi-agent collaboration benchmarks demonstrate that our method outperforms baselines. These findings highlight the importance of coalition-level credit assignment and cooperative games for advancing multi-agent learning.

## 1 Introduction

Cooperative Multi-Agent Reinforcement Learning (MARL) aims to train a group of agents to jointly maximize a shared objective in a common environment (Panait & Luke, 2005). Such a paradigm has shown great potential in a wide range of applications (Hu et al., 2023), including autonomous driving platoons (Shalev-Shwartz et al., 2016), multi-robot systems (Busoniu et al., 2008), and large-scale network control (Ma et al., 2024). A key challenge in MARL is how to effectively coordinate decentralized agents so that they can learn global strategies that maximize the global return (Oliehoek et al., 2008; Lowe et al., 2017).

Recent advances in policy-gradient algorithms have significantly improved stability and scalability in multi-agent learning. Among these, MAPPO (Yu et al., 2022), a multi-agent extension of PPO (Schulman et al., 2017), has become a state-of-the-art baseline for cooperative MARL. Building upon this, HAPPO and HATRPO (Kuba et al., 2021; Zhong et al., 2023) introduced sequential agent-wise updates to further stabilize learning, achieving superior performance across various benchmarks.

However, these methods typically share the same global advantage value across agents, which can result in suboptimal updates. This is primarily due to the synchronous nature of policy updates, where shared credit fails to distinguish individual contributions and may hinder cooperation. Such issues are often attributed to the Relative Overgeneralization (RO) problem. To mitigate this, several approaches have explored more refined credit assignment techniques. Value-based methods like VDN (Sunehag et al., 2017) and QMIX (Rashid et al., 2018), QTRAN (Son et al., 2019), QPLEX (Wang et al., 2020b) and policy-gradient methods like LICA (Zhou et al., 2020), COMA (Foerster et al., 2018), VDAC (Su et al., 2021), and FACMAC (Peng et al., 2021) assign credit from an individual perspective and have improved coordination efficiency (Wang et al., 2022b; 2020c). Additionally, DOP (Wang et al., 2020d) has tackled the exploration challenge from a maximum entropy perspective.

Despite their success, these methods focus exclusively on either global or individual perspectives. Between these extremes lies an underexplored middle ground: coalitional granularity, where credits are evaluated and allocated at the level of agent subsets (i.e., coalitions $C \subseteq N$). To address this gap, recent works have introduced Shapley value-based credit assignment from cooperative game theory into policy gradient methods (Wang et al., 2020a; Li et al., 2021; Wang et al., 2022a). While these approaches provide theoretically grounded individual attributions, they often lack interpretability in the context of multi-agent policy updates and rely on rigid baselines (e.g., no-op or zero actions), which reduce flexibility. Furthermore, many other meaningful cooperative game solutions remain unexplored in MARL.

In this paper, we propose **Core Advantage Decomposition (CORA)**, a novel credit assignment framework for multi-agent policy gradient methods. CORA estimates *coalitional advantages* by evaluating the marginal contributions of coalitions to the global return and decomposes credit using the *core solution* from cooperative game theory. This ensures *coalitional rationality* and preserves beneficial exploratory behaviors. To improve scalability, CORA employs *random coalition sampling* for efficient approximation.

The main contributions of this paper are threefold:

- Coalition-level credit assignment. We propose a novel coalitional advantage formulation and allocate credits via the strong $\epsilon$-Core, ensuring both global consistency and coalition rationality.
- Theoretical guarantees. We provide policy-improvement lower bounds at the coalition level, showing that CORA systematically reinforces beneficial coalitions. The coalitions with high potential advantage values will receive higher advantage values to promote collaborative strategy optimization.
- Practical effectiveness. We develop an efficient sampling approximation and demonstrate consistent performance gains across diverse MARL benchmarks, including matrix games, differential games, VMAS, SMAC, Google Research Football, and Multi-Agent MuJoCo.

## 2 RELATED WORK

This section provides an overview of key research areas relevant to our work, including traditional value decomposition methods, policy gradient methods.

### 2.1 VALUE DECOMPOSITION METHODS

Value decomposition methods aim to decompose the global value function in MARL into individual contributions from each agent, thereby facilitating decentralized learning. Value-Decomposition Networks (VDN) (Sunehag et al., 2017) is a pioneering approach that splits the joint action-value function into simpler, agent-specific value functions. This decomposition significantly reduces the complexity of multi-agent learning and allows for decentralized execution. QMIX (Rashid et al., 2018), an extension of VDN, introduces a monotonic mixing function that ensures the global Q-value is a monotonic combination of individual agent Q-values.

### 2.2 MULTI-AGENT POLICY GRADIENT METHODS

Policy gradient methods, particularly MAPPO (Multi-Agent Proximal Policy Optimization) (Yu et al., 2022), have become the dominant paradigm in MARL. MAPPO has shown significant performance improvements over earlier methods, such as COMA (Counterfactual Multi-Agent Policy Gradients) (Foerster et al., 2018) and MADDPG (Multi-Agent Deep Deterministic Policy Gradient) (Lowe et al., 2017). Attention-based credit assignment methods such as ATA (She et al., 2022) and spatiotemporal decomposition approaches such as STAS (Chen et al., 2024) improve multi-agent coordination by learning expressive representations of agent–time or space–time interactions. In contrast, our method formulates advantage decomposition as an $\epsilon$-core problem, ensuring that the allocated per-agent advantages satisfy coalition constraints for all sampled coalitions. Thus, CORA provides a complementary perspective: rather than learning a decomposition implicitly, it enforces a principled game-theoretic structure that guarantees consistency across coalitions.

### 2.3 Credit Assignment based on Shapley value

Shapley-based methods in MARL, integrating cooperative game theory, address the credit assignment problem by fairly distributing rewards based on each agent's contribution. Early work, such as SQDDPG (Wang et al., 2020a), uses the Shapley value in Q-learning and DDPG for continuous action spaces to calculate each agent's marginal contribution.

A more recent advancement, Shapley Counterfactual Credit Assignment (SCCA) (Li et al., 2021), refines credit assignment by considering counterfactual scenarios, improving accuracy and stability. However, SCCA faces computational challenges in multi-agent settings. SHAQ-learning (Wang et al., 2022a) also integrates Shapley values into Q-learning, enhancing stability and fairness in cooperative tasks, but it struggles with scalability and efficiency.

### 2.4 Cooperative Game Theory and the Core

Cooperative game theory, traditionally used in economics (Driessen, 2013), is also applied in MARL for credit assignment. Recent works (Jia et al., 2019), (Ghorbani & Zou, 2019), and (Sim et al., 2020) have adopted the Shapley value for data valuation and reward allocation. In federated learning, (Chaudhury et al., 2022) and (Donahue & Kleinberg, 2021) applied cooperative game theory to fairness and stability.

The core (Driessen, 2013), another key concept in cooperative game theory, guarantees stability by ensuring no coalition of agents can improve their outcome by deviating from the allocation. While the Shapley value has been used for fair reward distribution in MARL, traditional methods often rely on a fixed baseline, limiting their applicability in dynamic environments. Additionally, they do not address interference from high-risk explorations in cooperative MARL.

## 3 Background

This section provides an overview of the foundational concepts and challenges in MARL, focusing on policy gradient methods and the credit assignment methods.

### 3.1 Problem Formulation

In cooperative multi-agent reinforcement learning, a group of agents works together to maximize a shared return within a common environment (Panait & Luke, 2005; Kuba et al., 2021). This setting can be formalized as a Markov game (Littman, 1994; Kuba et al., 2021; Zhao et al., 2024) defined by the tuple $\mathcal{G} = \langle N, S, A, \mathbb{P}, r, \gamma \rangle$, where $N = \{1, \dots, n\}$ is the set of agents, $S$ is the state space, $A = \prod_{i \in N} A_i$ is the joint action space, with $A_i$ being the action space of agent $i$, $\mathbb{P} : S \times A \times S \to [0, 1]$ is the transition function, $r : S \times A \to \mathbb{R}$ is the reward function, and $\gamma \in [0, 1)$ is the discount factor. At each time $t \in \mathbb{N}$, each agent $i$ observes the full state $s^t$, and selects an action $a_i^t \in A_i$ drawn from its policy $\pi_i(\cdot | s^t)$. The joint action $a^t = (a_1^t, \dots, a_n^t)$ leads to the next state $s^{t+1} \sim \mathbb{P}(s^{t+1} | s^t, a^t)$ and generates a common reward $r^t = r(s^t, a^t)$ for all agents. The agents aim to updated their policies that maximize the shared expected cumulative reward:

$$\max_{\pi} J(\pi) = \mathbb{E}_{s, a \sim \pi, \mathbb{P}} \left[ \sum_{t=0}^{\infty} \gamma^t r(s_t, a_t) \right]. \tag{1}$$

Under the centralized training with decentralized execution (CTDE) paradigm Oliehoek et al. (2008); Lowe et al. (2017); Yu et al. (2022), each agent $i$ is trained with global information and execute using only local observation $o_i = O_i(s) \in \mathcal{O}_i$. A central component in training process is the global state value function $V(s)$ (the global state-action value function $Q(s, a)$), estimating the expected return from state $s$ (after taking joint action $a$). Denoting the advantage $A(s^t, a^t) = Q(s^t, a^t) - V(s^t)$ with GAE estimator

$$A_{GAE}^t = \sum_{l=0}^{\infty} (\gamma \lambda)^l \delta_{t+l} \tag{2}$$

where $\delta_t$ denotes the TD error $\delta_t = r_t + \gamma V(s_{t+1}) - V(s_t)$, a standard multi-agent policy gradient for agent $i$ is

$$\nabla_{\phi_i} J = \mathbb{E}\left[\nabla_{\phi_i} \log \pi_i(a_i|s) A_i(s,a)\right], \tag{3}$$

where individual advantage $A_i(s,a)$ is the per-agent credit signal. Sharing the same advantage $A(s,a)$ across agents is simple and stable, but it fails to capture heterogeneous contributions of different agents, leading to inefficient credit assignment and slower convergence.

Throughout this paper, we focus on multi-agent credit assignment via advantage decomposition for policy-gradient methods, using it to drive policy updates that strengthen effective coalitional collaboration.

## 3.2 SHARING ADVANTAGE

Many credit assignment methods such as COMA Foerster et al. (2018), VDN Sunehag et al. (2017), QMIX Rashid et al. (2018), and LICA Zhou et al. (2020) assign advantage or value from individual or marginal perspective. In this paper, besides the global advantage, we also consider the coalitional advantage for each coalition of agents. Let $N = \{1, \ldots, n\}$ denote the set of all agents. For a given sample $(s,a)$ where $s = (s_1, \cdots, s_n)$ and $a = (a_1, \cdots, a_n)$, we evaluate the scenario where agents in coalition $C \subseteq N$ take actions $a_C$, while each agents $i \notin C$ execute a baseline action $\bar{a}_i$ or current policy $\pi_{N \setminus C}$.

Sharing the global advantage $A(s,a)$ among agents often leads to insufficient policy updates. This method incentivizes each agent to update its policy $\pi_i(a_i|s_i)$ to either approach action $a_i$ with $A(s,a) > 0$ or avoid those with $A(s,a) < 0$. Specifically, when an action $a$ with $Q(s,a) < V(s)$ is explored during training, all agents are penalized via $A(s,a) < 0$, and the policy $\pi_i(a_i|s_i)$ for each agent is updated to reduce its probability. This occurs even if a coalition $C$ could form a superior joint action $(a_C, a'_{N \setminus C})$ satisfying $Q(s, a_C, a'_{N \setminus C}) > V(s)$.

Moreover, consider the case where the executed action $a^*$ is already optimal. If agents in coalition $C$ explore a new action $a_C$ while others act optimally, and $Q(s, a_C, a^*_{N \setminus C}) < V(s)$, then the probability $\pi_i(a_i^*|s_i)$ for each agent $i \notin C$ is reduced due to $A(s,a) < 0$, destabilizing the probability distribution over the optimal action $a^*$.

In summary, the value of coalition actions can be further exploited. Agents with greater potential, such as those belonging to a coalition $C$ where $Q(s, a_C, \bar{a}_{N \setminus C}) \ll V(s)$ or $\mathbb{E}_{a_{N \setminus C} \sim \pi_{N \setminus C}}[Q(s, a_C, a_{N \setminus C})] \ll V(s)$, should receive larger advantage values to encourage the action $(a_C, \bar{a}_{N \setminus C})$.

# 4 CORE ADVANTAGE DECOMPOSITION FOR MULTI-AGENT POLICY GRADIENTS

In this section, we evaluate the advantage of coalition actions and propose an advantage decomposition algorithm.

## 4.1 COALITIONAL ADVANTAGE

Consider a global value function $Q(s,a)$, which describes the return of the joint action $a$ in state $s$. The advantage of coalition $C$, denoted as $A_C(s, a_C)$, is defined as:

$$A_C(s, a_C) = \mathbb{E}_{a_{N \setminus C} \sim \pi_{N \setminus C}}[Q(s, a_C, a_{N \setminus C})] - V(s), \tag{4}$$

where the first term $\mathbb{E}_{a_{N \setminus C} \sim \pi_{N \setminus C}}[Q(s, a_C, a_{N \setminus C})]$ represents the expected return when coalition $C$ takes the sampled action $a_C$, and the other agents $i \notin C$ follow the current strategy $\pi_{N \setminus C}$. Subtracting the baseline value $V(s)$ gives the advantage of coalition $C$ taking action $a_C$ alone. Incidentally, the global value naturally satisfies $A_N(s,a) = Q(s,a) - V(s) = A(s,a)$. By defining the advantage in this way, we can clearly quantify the contribution of each coalition action $a_C$ to the team.

Besides the definition $A_C(s, a_C)$, we can also consider defining it as:

$$A_C(s, a_C) = Q(s, a_C, \bar{a}_{N \setminus C}) - V(s) \tag{5}$$

where $\bar{a}_i$ represents a baseline action. The baseline action $\bar{a}$ provides a reference for evaluating coalition values. In our experiments, we mainly consider the most probable action as the baseline action. This is because, regardless of whether a discrete softmax policy or a continuous Gaussian policy is used, the most probable action is typically chosen during evaluation or execution. During training, however, actions are sampled from the probability distribution to encourage exploration. Specifically: (i) For discrete actions: $\bar{a}_i = \arg\max_{a_i} \pi_{\theta_i}(a_i|s_i)$, while training samples are drawn from $\pi_i(\cdot|s_i)$; (ii) For continuous actions: $\bar{a}_i = \mu_{\theta_i}(s_i)$. For example, a Gaussian policy outputs $(\mu_i, \sigma_i)$, with training samples $a_i \sim \mathcal{N}(\mu_i, \sigma_i)$, while the baseline action uses $\mu_{\theta_i}(s_i)$.

## 4.2 Advantage Decomposition

The next problem we need to solve is how to allocate advantage $A_i(s, a)$ to each agent $i \in N$ based on $2^n$ advantage values $A_C(s, a_C)$ (for each $C \subseteq N$). Intuitively, if a coalition action $a_C$ yields a high advantage value $A_C(s, a_C)$, the total advantage assigned to the agents in that coalition should not be too small. Formally, we require

$$\sum_{i \in C} A_i(s, a) \geq A_C(s, a_C) - \epsilon. \tag{6}$$

This allocation principle aligns with coalitional rationality in cooperative game theory. If coalition actions $(a_C, \pi_{N \setminus C})$ are promising, it is beneficial to incentivize each $a_i$ ($i \in C$) to adjust its policy distribution, thereby encouraging exploration of this action in the future.

Additionally, it is essential to ensure that $\sum_{i \in N} A_i(s, a) = A_N(s, a) = A(s, a)$, which is known as effectiveness in cooperative game theory and is also widely adopted as a guiding principle in value decomposition methods. For convenience, given current state $s$ and action $a$, we denote the advantage value of agent $i$, $A_i(s, a)$, simply as $A_i$.

This form coincides with the classic solution Strong $\epsilon$-Core of cooperative game theory Driessen (2013):

$$\text{Core}_\epsilon(N, A_C) = \Big\{ (A_1, \cdots, A_n) \in \mathbb{R}^n \mid \sum_{i \in N} A_i = A_N(s, a),$$
$$\sum_{i \in C} A_i \geq A_C(s, a_C) - \epsilon, \text{for } \forall C \subseteq N \Big\}, \tag{7}$$

where $\epsilon \geq 0$ is a non-negative parameter that allows for a small deviation from the ideal condition.

Generally, the $\epsilon$-core may admit infinitely many solutions, but not all of them are desirable. In particular, some allocations satisfying coalition rationality may place all credit on a single agent, leaving others without effective incentives. To avoid such imbalanced solutions, we introduce an additional objective that penalizes large deviations from the uniform allocation. Specifically, we minimize the variance of credits among agents, leading to the quadratic program:

$$\underset{\epsilon \geq 0, A_1, \ldots, A_n}{\text{minimize}} \quad \epsilon + \lambda \sum_{i \in N} \left( A_i - \frac{1}{|N|} A_N(s, a) \right)^2,$$
$$\text{subject to: } \sum_{i \in N} A_i = A_N(s, a), \tag{8}$$
$$\sum_{i \in C} A_i \geq A_C(s, a_C) - \epsilon, \forall C \subseteq N.$$

This formulation ensures a more balanced allocation while respecting coalition rationality. In detail, $\mathbb{E}_{a_{N \setminus C}}[Q(s, a)]$ can be estimated using Monte Carlo sampling, approximately given by $\frac{1}{|K|} \sum_{k \in K} Q(s, a_C, a_{N \setminus C}^k)$ where $K$ is the set of sampled trajectories, and $a_{N \setminus C}^k$ represents the action taken by the agents in $N \setminus C$ during the $k$-th trajectory. The diagram and pseudocode are shown as Figure 7 and Algorithm 1. In summary, our framework requires two critics, $Q(s, a)$ and $V(s)$, both updated using temporal-difference (TD) errors. The value critic $V$ is employed for generalized advantage estimation (GAE), which stabilizes policy updates; in addition, the global advantage (for grand coalition $N$) $A_N(s, a)$ is also estimated based on GAE. The state-action value critic $Q$, on the other hand, is responsible for allocating the advantage $A_i$.

# 5 THEORETICAL ANALYSIS

In this section, we provide some theoretical analysis and approximate methods based on the designed CORA advantage value.

**Theorem 1.** *Under compatible approximation and a natural policy gradient (NPG) step, for small step size $\alpha > 0$,*

$$\Delta \log \pi_i(a_i \mid s) \approx \alpha A_i,$$

$$\Delta \log \pi(a \mid s) = \sum_{i=1}^{n} \Delta \log \pi_i(a_i \mid s) \approx \alpha A_N,$$

$$\Delta \log \pi_C(a_C \mid s) = \sum_{i \in C} \Delta \log \pi_i(a_i \mid s) \approx \alpha \sum_{i \in C} A_i.$$

**Theorem 2.** *Consider one NPG step $\phi_i' = \phi_i + \alpha F_i^{-1} g_i$ with $g_i := \mathbb{E}[\psi_i A_i]$, $\psi_i := \nabla_{\phi_i} \log \pi_i(a_i \mid s)$, $F_i := \mathbb{E}[\psi_i \psi_i^\top]$, and step size $\alpha > 0$. Assume for each agent $i$ that $\log \pi_i(\cdot \mid s; \phi_i)$ is twice continuously differentiable and its Hessian is uniformly bounded on the line segment between $\phi_i$ and $\phi_i'$:*

$$\left\| \nabla_{\phi_i}^2 \log \pi_i(a_i \mid s; \xi_i) \right\|_{\mathrm{op}} \leq L_i \quad \text{for all } \xi_i \in [\phi_i, \phi_i'].$$

*Then for any coalition $C \subseteq N$ and any sampled $(s, a)$,*

$$\Delta \log \pi_C(a_C \mid s) \geq \alpha \sum_{i \in C} A_i - \frac{\alpha^2}{2} \sum_{i \in C} L_i \left\| F_i^{-1} g_i \right\|_2^2. \tag{9}$$

*If, in addition, the strong $\epsilon$-Core constraints hold, $\sum_{i \in C} A_i \geq A_C(s, a_C) - \epsilon$, then*

$$\Delta \log \pi_C(a_C \mid s) \geq \alpha \Big( A_C(s, a_C) - \epsilon \Big) - \frac{\alpha^2}{2} \sum_{i \in C} L_i \left\| F_i^{-1} g_i \right\|_2^2. \tag{10}$$

**Theorem 3.** *Let $C^\star \in \arg\max_{C \subseteq N} A_C(s, a_C)$. Under the strong $\epsilon$-Core, $\sum_{i \notin C^\star} A_i \leq \epsilon$ and thus $\Delta \log \pi_{N \setminus C^\star}(a_{N \setminus C^\star} \mid s) \lesssim \alpha \epsilon$, while $\Delta \log \pi_{C^\star}(a_{C^\star} \mid s) \gtrsim \alpha (A_{C^\star} - \epsilon)$.*

Solving the quadratic programming problem (8) requires $2^{|N|}$ inferences of the value network to obtain $\mathbb{E}_{a_{N \setminus C}}[Q(s, a_C, a_{N \setminus C})]$ or $Q(s, a_C, \bar{a}_{N \setminus C})$ for each coalition $C \subseteq N$. This may results in significant computational overhead for large-scale problems. To address this issue, our method randomly samples a relatively small number of coalitions $\mathcal{C} = \{C_1, C_2, \cdots, C_m\}$ and computes the desired solution satisfing the constraints of these coalitions, resulting in the quadratic programming 19. Theorem 4 shows that its approximation error can be controlled by the sample size $m$.

**Theorem 4.** *Given a distribution $\mathcal{P}$ over $2^N$, and $\delta$, $\Delta > 0$, solving the programming (19) over $O((n + 2 + \log(1/\Delta))/\delta^2)$ coalitions sampled from $\mathcal{P}$ gives an allocation vector in the $\delta$-probable core with probability $1 - \Delta$.*

The above proof is for the general form of $A_C(s, a_C)$. For a more rigorous approach, we use the detailed definition $A_C(s, a_C) = \mathbb{E}_{a_{N \setminus C} \sim \pi_{N \setminus C}}[Q(s, a_C, a_{N \setminus C})] - V(s)$. Using the PPO/TRPO framework, we arrive at the following conclusions.

**Theorem 5.** *Given a factored joint policy $\pi(a \mid s) = \prod_{i \in N} \pi_i(a_i \mid s)$ and the CORA advantage allocation satisfying the coalition constraint*

$$\sum_{j \in C} A_j(s, a) \geq A_C(s, a_C) - \epsilon, \qquad \forall C \subseteq N,$$

*the following hold for the trust-region penalized policy update with parameter $\eta > 0$.*

**(1) Individual improvement lower bound.** *Assume $m_i \leq A_i(s, a) \leq M_i$ with $R_i = M_i - m_i$. Then each agent satisfies*

$$\Delta \log \pi_i(a_i \mid s) \geq \eta(A_i(s, a) - \epsilon) - \frac{\eta^2 R_i^2}{8}.$$

**(2) Coalition improvement lower bound.** *For any coalition $C \subseteq N$,*

$$\Delta \log \pi_C(a_C \mid s) \geq \eta\big(A_C(s, a_C) - (1 + |C|)\epsilon\big) - \sum_{i \in C} \frac{\eta^2 R_i^2}{8}.$$

*These bounds imply that CORA protects high-value coalitions by enforcing a guaranteed positive advantage contribution even when the global joint advantage $A_N(s, a)$ is weak or negative.*

## 6 EXPERIMENTS

We evaluate the CORA method across several multi-agent environments, including matrix games, differential games, the VMAS simulator Bettini et al. (2022), the Multi-Agent MuJoCo (MaMujoco) environment de Lazcano et al. (2024); Kuba et al. (2021), and the Starcraft Multi-Agent Challenge (SMAC) environment Samvelyan et al. (2019); Hu et al. (2023).

### 6.1 MATRIX GAMES

In this section, we construct two types of matrix-style cooperative game environments to evaluate the fundamental performance of different algorithms.

**Matrix Team Game (MTG)**: In this environment, agents receive a shared reward at each time step based on their joint action, determined by a randomly generated reward matrix. Each element of the matrix is uniformly sampled from the interval $[-10, 20]$. The game proceeds for 10 steps per episode. Each agent observes a global one-hot encoded state indicating the current step number, allowing them to learn time-dependent coordination strategies.

**Multi-Peak Matrix Team Game**: To further evaluate each algorithm's ability to optimize cooperative strategies in environments with multiple local optima, we extend MTG to design a more challenging setting. The matrix is filled with background noise in the range $[-10, 0]$, overlaid with multiple reward peaks. Among them, one peak is the global optimum (highest value), while the rest are local optima. Actions deviating from peak combinations incur heavy penalties due to the negative background. This setting is designed to test whether algorithms can escape suboptimal solutions and discover globally coordinated strategies.

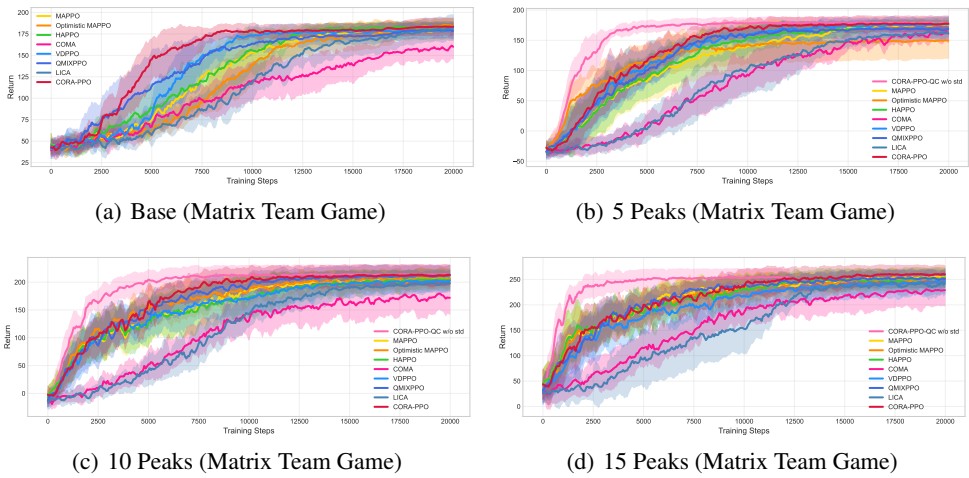

(a) Base (Matrix Team Game)  (b) 5 Peaks (Matrix Team Game)

(c) 10 Peaks (Matrix Team Game)  (d) 15 Peaks (Matrix Team Game)

Figure 1: Training performance on Matrix Team Game and its Multi-Peak variants with 5, 10, and 15 reward peaks.

As shown in Figure 1, CORA exhibits faster convergence and higher returns compared to the baseline algorithms, demonstrating superior coordination and learning efficiency in this simple and general environment. As a comparison, we also implemented a quadratic critic that directly parameter-

izes the joint action value as a quadratic form:

$$Q(s, a) = b(s) + \sum_i \langle u_i(s), a_i \rangle + \sum_{i<j} a_i^\top W_{ij}(s) a_j,$$

where $a_i$ is the one-hot or probabilistic action vector of agent $i$. This representation allows us to evaluate coalition values in closed form with baseline $a_i \sim \pi_i(a_i|s)$ for $i \notin C$, e.g.,

$$Q_C(s, a_C) = \mathbb{E}_{a_{N \setminus C} \sim \pi_{N \setminus C}} \big[ Q(s, a_C, a_{N \setminus C}) \big],$$

by simply replacing the action inputs of non-coalition agents with their policy distributions. The results, shown as CORA-PPO-QC, confirm this gap and highlight the stability advantage of CORA.

## 6.2 DIFFERENTIAL GAMES

To demonstrate the learning process, we designed a 2D differential game environment (similar to (Wei & Luke, 2016)). Each agent selects an action $x_1, x_2 \in [-5, 5]$ at every step. The reward function $R(x_1, x_2)$ is composed of a sum of several two-dimensional Gaussian potential fields, defined as:

$$R(x_1, x_2) = \sum_{i=1}^n h_i \cdot \exp\left( -\frac{(x_1 - c_{x_i})^2 + (x_2 - c_{y_i})^2}{\sigma_i^2} \right) \tag{11}$$

Here, $n$ is the number of fields, $(c_{x_i}, c_{y_i})$ is the center of the $i$-th potential field, $h_i \in [5, 10]$ indicates the peak height of the potential field, and $\sigma_i \in [1, 2]$ controls its spread. This setup results in an environment with multiple local optima, presenting significant strategy exploration and learning challenges for MARL algorithms. The environment state itself does not evolve and can be regarded as a repeated single-step game. Key parameters like location, height, width of potential fields are set by a random seed.

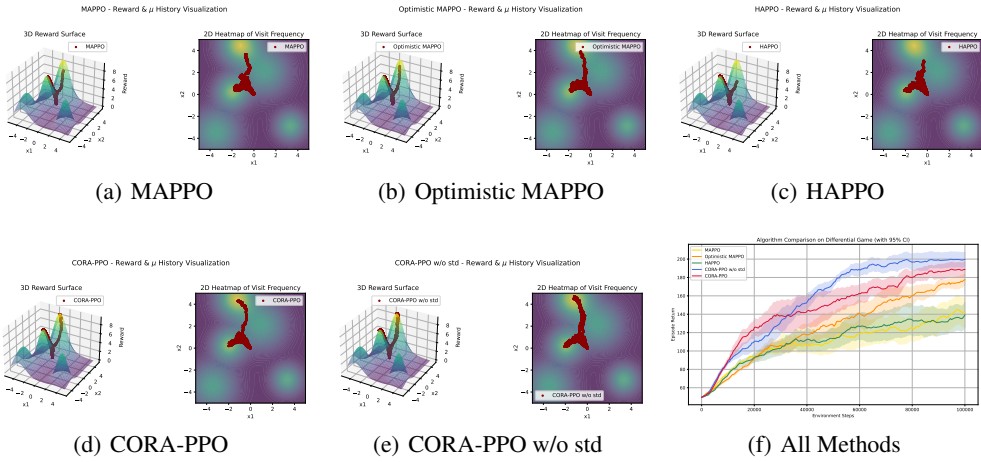

(a) MAPPO  (b) Optimistic MAPPO  (c) HAPPO

(d) CORA-PPO  (e) CORA-PPO w/o std  (f) All Methods

Figure 2: The reward and learning trajectories of various algorithms in the differential game scenario ($\mu$ in Gaussian strategy).

Figure 2(f) shows the performance of MAPPO, HAPPO, CORA-PPO, CORA-PPO without std, and Optimistic MAPPO in this environment. CORA-PPO demonstrates the best learning speed and performance, and CORA-PPO without std outperforms other algorithms. We believe this is because the std term somewhat suppresses agent exploration. Since the differential game has multiple local optima, the std term constrains exploration across these optima. Furthermore, thanks to the theory of Optimistic Q-learning, Optimistic MAPPO also outperforms both HAPPO and MAPPO in this environment.

The detailed learning trajectories are visualized in Figure 2, which illustrate the learning trajectories of various methods during training (through the mean $\mu_i$ in the Gaussian policy $N(\mu_i, \sigma_i)$). It is clearly visible that the CORA-PPO series effectively promotes agents to learn optimal cooperative strategies (reaching the peak in the 3D Reward Surface; reaching the brightest point in the 2D Heatmap).

### 6.3 VMAS

VMAS (Vectorized Multi-Agent Simulator) is a PyTorch-based vectorized multi-agent simulator designed for efficient multi-agent reinforcement learning benchmarking Bettini et al. (2022; 2024); Bou et al. (2023). It provides a range of challenging multi-agent scenarios, and utilizes GPU acceleration, making it suitable for large-scale MARL training. We selected the following scenarios for testing: **Multi-Give-Way**: Four agents must coordinate to cross a shared corridor by giving way to each other to reach their respective goals. **Give-Way**: Two agents are placed in a narrow corridor with goals on opposite ends. Success requires one agent to yield and allow the other to pass first, reflecting asymmetric cooperative behavior. **Navigation**: Agents are randomly initialized and must navigate to their own goals while avoiding collisions. These tasks require strong coordination and implicit role assignment.

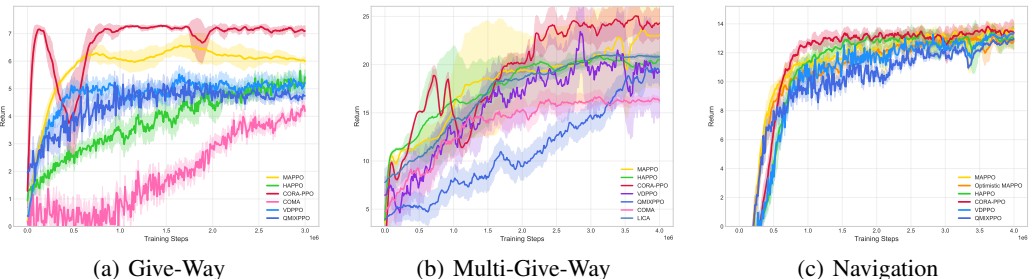

(a) Give-Way         (b) Multi-Give-Way         (c) Navigation

Figure 3: Training performance on the VMAS scenarios.

As shown in Figure 3, CORA achieves higher returns and more stable performance compared to the other algorithms.

### 6.4 MULTI-AGENT MUJOCO

To demonstrate the effectiveness of CORA in continuous control tasks, we conducted experiments on the popular benchmark Multi-Agent MuJoCo (MA-MuJoCo) Kuba et al. (2021), using its latest version, MaMuJoCo-v5 de Lazcano et al. (2024).

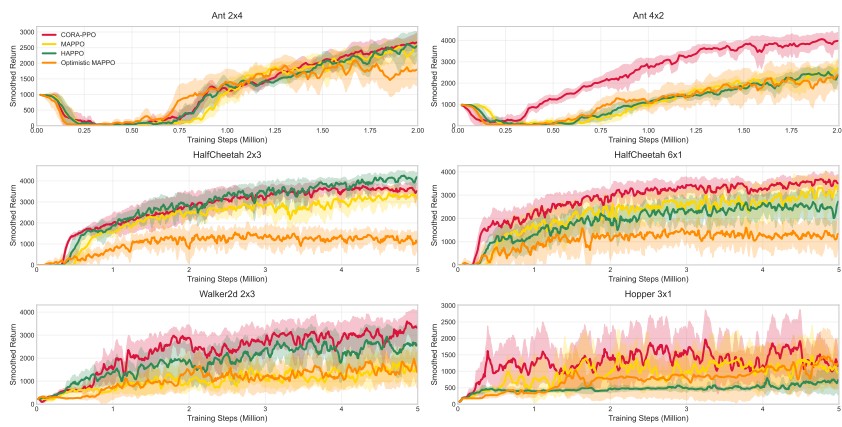

Figure 4: Training performance in the Multi-Agent MuJoCo (MaMuJoCo-v5) scenario.

As shown in Figure 4, CORA-PPO achieves state-of-the-art performance across multiple scenarios. Except for the *HalfCheetah 2x3* task where HAPPO slightly outperforms, CORA-PPO demonstrates superior results in the *Ant 4x2*, *HalfCheetah 6x1*, *Walker2d 2x3*, and *Hopper 3x1* tasks. These results highlight the effectiveness of CORA in handling diverse and challenging multi-agent continuous control environments.

## 6.5 STARCRAFT MULTI-AGENT CHALLENGE (SMAC)

We further validate the scalability and cooperation capability of CORA-PPO on the StarCraft Multi-Agent Challenge (SMAC). All experiments are conducted using SC2 version 4.10 under decentralized execution with global team reward, following standard protocols. Results are averaged over 8 runs with 95% confidence intervals.

Five representative maps of different cooperative difficulty are selected: 3s_vs_5z, 8m, 2s_vs_1sc, 2s3z, and 3m. These maps involve heterogeneous team sizes, spatial complexity, and micro-control demands, creating challenges in credit assignment and coordinated tactics.

Figures 5(a)–5(e) show that CORA-PPO consistently achieves higher win rates and faster convergence than MAPPO and HAPPO. In the more demanding maps (*3s_vs_5z* and *2s_vs_1sc*), CORA-PPO notably enhances cooperative unit control and improves asymptotic performance, demonstrating its robustness in partial observability and high-interaction combat.

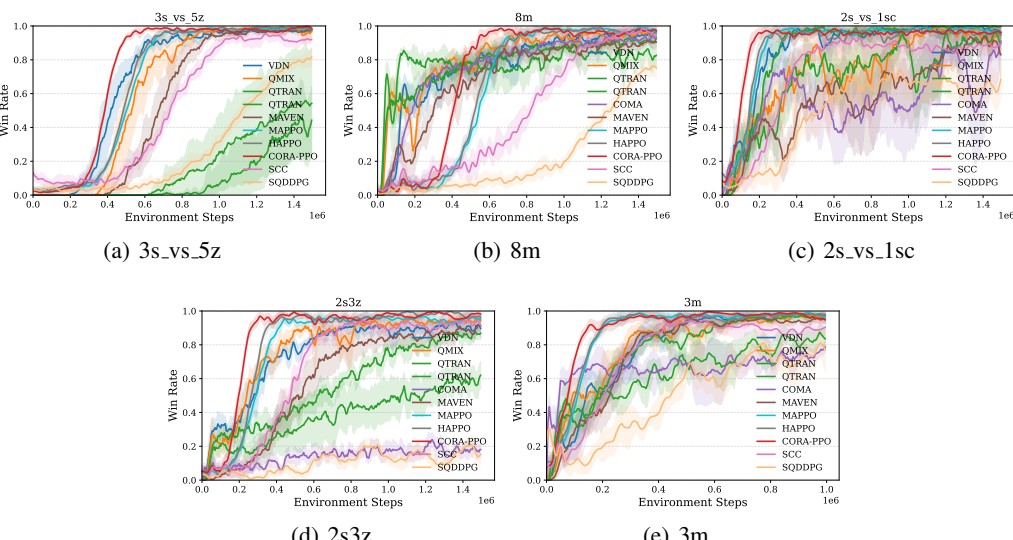

(a) 3s_vs_5z          (b) 8m          (c) 2s_vs_1sc

(d) 2s3z          (e) 3m

Figure 5: Win rate comparison across SMAC scenarios.

## 6.6 GOOGLE RESEARCH FOOTBALL

We further evaluate CORA-PPO on the Google Research Football (GRF) benchmark. Results are averaged over 8 random seeds with 95% confidence intervals.

We consider three representative cooperative tasks: *3 vs 1 with keeper* (3 agents), *counterattack_easy*, and *counterattack_hard*. As shown in Figures 6(a)–6(c), CORA-PPO achieves higher returns and more stable training, indicating that coalition-aware credit assignment improves cooperative decision-making under sparse and delayed rewards.

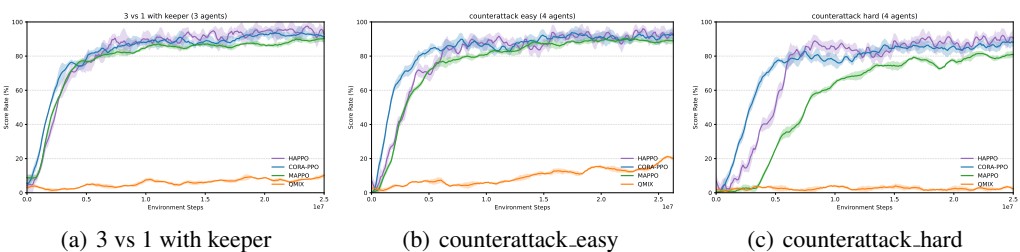

(a) 3 vs 1 with keeper          (b) counterattack_easy          (c) counterattack_hard

Figure 6: Performance comparison on GRF scenarios.

ETHICS STATEMENT

We have read and will adhere to the ICLR Code of Ethics. This work does not involve human subjects, personally identifiable information, or sensitive attributes. No new datasets with personal data are collected. Experiments are conducted in standard public benchmarks under their respective licenses.

Potential negative societal impacts: our method could be used to optimize multi-agent coordination in safety-critical or competitive scenarios. To mitigate risks, we (i) avoid claims beyond measured settings; (ii) release only research artifacts necessary to reproduce results; and (iii) encourage deployment-time safeguards (e.g., monitoring, intervention policies). We are unaware of legal compliance issues specific to the presented experiments.

Conflicts of interest: none declared.

REPRODUCIBILITY STATEMENT

We take the following steps to support reproducibility. (1) **Algorithm details.** CORA's objective and constraints are specified in Sec. 4.2 (Eq. 8, 19); the policy-gradient estimator and baselines are defined in Sec. 3. (2) **Implementation.** Pseudocode is provided in 1. (3) **Hyperparameters.** Complete training hyperparameters per environment are listed in Table 1 (actor/critic learning rates, $\gamma$, GAE $\lambda$, PPO clip, parallel envs, epochs). (4) **Environments & seeds.** We describe matrix/differential games, VMAS, and Multi-Agent MuJoCo settings in Sec. 6 and Appendix, including action/state spaces, reward definitions, and episode lengths. We run 5 random seeds for both environment and algorithm initializations and report mean with $95\%$ confidence intervals. (5) **Code & artifacts.** Anonymized code and configuration files (including environment wrappers and plotting scripts) will be provided in the supplementary materials; instructions include exact package versions, and commands to reproduce all figures. (6) **Ablations.** We report the effect of coalition sample size and the variance regularizer in Appendix (Fig. 8, 9).

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

# A   APPENDIX

In this appendix, we provide detailed statements of the theorems, lemmas, and their corresponding proofs presented in the main text.

## A.1   PRELIMINARIES AND NOTATION

The actor parameters are $\phi = (\phi_1, \dots, \phi_n)$ and the factored joint policy $\pi_\phi(a \mid s) = \prod_{i=1}^n \pi_i(a_i \mid s; \phi_i)$. Define score features and per-agent Fisher matrices

$$\psi_i(s, a_i) := \nabla_{\phi_i} \log \pi_i(a_i \mid s), \qquad F_i := \mathbb{E}[\psi_i \psi_i^\top], \qquad F = \operatorname{diag}(F_1, \dots, F_n) \succeq 0.$$

The NPG step is

$$\phi_i' = \phi_i + \alpha\, F_i^{-1} g_i, \qquad g_i := \mathbb{E}[\psi_i A_i], \tag{12}$$

for step size $\alpha > 0$. Global advantage $A_N(s, a) := Q(s, a) - V(s)$ and coalitional advantage $A_C(s, a_C)$. A credit allocation $\{A_i\}_{i \in N}$ satisfies the strong $\epsilon$-Core if

$$\sum_{i \in N} A_i = A_N, \qquad \sum_{i \in C} A_i \geq A_C(s, a_C) - \epsilon, \ \forall C \subseteq N. \tag{13}$$

## A.2   COMPATIBLE FUNCTION APPROXIMATION

**Definition 6** (Compatible approximation)**.** *For agent $i$, consider $\mathcal{S}_i = \{w_i^\top \psi_i : w_i \in \mathbb{R}^{d_i}\}$. We say $A_i$ is compatibly representable if*

$$w_i^\star = \arg\min_w \ \mathbb{E}\big[(A_i - w^\top \psi_i)^2\big]$$

*exists and satisfies the normal equation $\mathbb{E}[\psi_i A_i] = \mathbb{E}[\psi_i \psi_i^\top] w_i^\star = F_i w_i^\star$.*

**Lemma 7.** *With $g_i = \mathbb{E}[\psi_i A_i]$, the NPG step equation 12 gives $\phi_i' - \phi_i = \alpha\, F_i^{-1} g_i = \alpha\, w_i^\star$.*

*Proof.* From $F_i w_i^\star = g_i$, left-multiply by $F_i^{-1}$ to obtain $w_i^\star = F_i^{-1} g_i$. Substitute into equation 12. $\square$

**Lemma 8.** *For small $\alpha$, the Taylor expansion yields*

$$\Delta \log \pi_i(a_i \mid s) := \log \pi_i'(a_i \mid s) - \log \pi_i(a_i \mid s) \approx \psi_i(s, a_i)^\top (\phi_i' - \phi_i).$$

*Proof.* Differentiate $\log \pi_i(a_i \mid s; \phi_i)$ at $\phi_i$ in direction $(\phi_i' - \phi_i)$. $\square$

## A.3   FIRST-ORDER CHANGES

**Theorem 1'.** *Under compatible approximation and equation 12,*

$$\Delta \log \pi_i(a_i \mid s) \approx \alpha\, A_i, \tag{14}$$

$$\Delta \log \pi(a \mid s) = \sum_{i=1}^n \Delta \log \pi_i(a_i \mid s) \approx \alpha\, A_N, \tag{15}$$

$$\Delta \log \pi_C(a_C \mid s) = \sum_{i \in C} \Delta \log \pi_i(a_i \mid s) \approx \alpha \sum_{i \in C} A_i. \tag{16}$$

*Proof.* By Lemma 8 and Lemma 7, $\Delta \log \pi_i \approx \psi_i^\top (\alpha w_i^\star) = \alpha w_i^{\star\top} \psi_i = \alpha A_i$, which proves equation 14. Because $\log \pi = \sum_i \log \pi_i$, summing equation 14 over $i$ and using $\sum_i A_i = A_{tot}(s, a)$ yields equation 15. Similarly, $\log \pi_C = \sum_{i \in C} \log \pi_i$ gives equation 16. $\square$

**Corollary 9.** *Using* $\Delta\pi(\cdot) \approx \pi(\cdot)\Delta\log\pi(\cdot)$, $\Delta\pi(a \mid s) \approx \alpha\,\pi(a \mid s)A_N$ *and* $\Delta\pi_C(a_C \mid s) \approx \alpha\,\pi_C(a_C \mid s)\sum_{i\in C} A_i$.

*Remark* 1 (If $A_i \notin \mathcal{S}_i$). All first-order relations remain valid with $A_i$ replaced by its $L^2$ projection onto $\mathcal{S}_i$. Operationally, NPG realizes this via $w_i^\star = F_i^{-1}\mathbb{E}[\psi_i A_i]$.

## A.4 COALITIONAL LOWER BOUNDS FROM THE STRONG $\epsilon$-CORE

**Theorem 2′.** *Consider one NPG step* $\phi_i' = \phi_i + \alpha\,F_i^{-1}g_i$ *with* $g_i := \mathbb{E}[\psi_i A_i]$, $\psi_i := \nabla_{\phi_i}\log\pi_i(a_i \mid s)$, $F_i := \mathbb{E}[\psi_i\psi_i^\top]$, *and step size* $\alpha > 0$. *Assume for each agent* $i$ *that* $\log\pi_i(\cdot \mid s; \phi_i)$ *is twice continuously differentiable and its Hessian is uniformly bounded on the line segment between* $\phi_i$ *and* $\phi_i'$:

$$\left\|\nabla^2_{\phi_i}\log\pi_i(a_i \mid s; \xi_i)\right\|_{\mathrm{op}} \;\leq\; L_i \quad \text{for all } \xi_i \in [\phi_i, \phi_i'].$$

*Then for any coalition* $C \subseteq N$ *and any sampled* $(s, a)$,

$$\Delta\log\pi_C(a_C \mid s) \;\geq\; \alpha\sum_{i\in C} A_i \;-\; \frac{\alpha^2}{2}\sum_{i\in C} L_i\left\|F_i^{-1}g_i\right\|_2^2. \tag{17}$$

*If, in addition, the strong* $\epsilon$-*Core constraints hold,* $\sum_{i\in C} A_i \geq A_C(s, a_C) - \epsilon$, *then*

$$\Delta\log\pi_C(a_C \mid s) \;\geq\; \alpha\Big(A_C(s, a_C) - \epsilon\Big) \;-\; \frac{\alpha^2}{2}\sum_{i\in C} L_i\left\|F_i^{-1}g_i\right\|_2^2. \tag{18}$$

*Proof.* For each $i$, apply the second-order Taylor expansion of $\log\pi_i(a_i \mid s; \phi_i)$ along the direction $\Delta\phi_i := \phi_i' - \phi_i$:

$$\Delta\log\pi_i(a_i \mid s) = \psi_i(s, a_i)^\top\Delta\phi_i + \frac{1}{2}\Delta\phi_i^\top\left(\nabla^2_{\phi_i}\log\pi_i(a_i \mid s; \xi_i)\right)\Delta\phi_i,$$

for some $\xi_i$ on the line segment between $\phi_i$ and $\phi_i'$. With $\Delta\phi_i = \alpha F_i^{-1}g_i$ and the operator-norm bound on the Hessian,

$$\Delta\log\pi_i(a_i \mid s) \;\geq\; \alpha\,\psi_i^\top F_i^{-1}g_i \;-\; \frac{\alpha^2}{2}\,L_i\left\|F_i^{-1}g_i\right\|_2^2.$$

By compatible approximation, $\psi_i^\top F_i^{-1}g_i = A_i$. Summing over $i \in C$ yields equation 17. Combining with the strong $\epsilon$-Core inequality gives equation 18. $\qquad\square$

## A.5 ADVANTAGE CONCENTRATION ON A MAXIMIZING COALITION

Let $C^\star \in \arg\max_{C\subseteq N} A_C(s, a_C)$, we have $A_{C^\star} \geq A_N$.

**Theorem 3′.** *Under equation 13:*

1. $\displaystyle\sum_{i\notin C^\star} A_i = A_N - \sum_{i\in C^\star} A_i \leq A_N - (A_{C^\star} - \epsilon) \leq \epsilon$, *hence* $\Delta\log\pi_{N\setminus C^\star}(a_{N\setminus C^\star} \mid s) \lesssim \alpha\,\epsilon$.

2. $\Delta\log\pi_{C^\star}(a_{C^\star} \mid s) \gtrsim \alpha\big(A_{C^\star} - \epsilon\big)$.

*Proof.* (1) From $\sum_{i\in C^\star} A_i \geq A_{C^\star} - \epsilon$ and $A_{C^\star} \geq A_N$, $\sum_{i\notin C^\star} A_i \leq \epsilon$; then apply equation 16 to the complement. (2) This is equation 15. $\qquad\square$

## A.6 THEOREM 4: APPROXIMATION WITH SAMPLED COALITIONS

The approximate quadratic programming problem mentioned in the main text is as follows.

$$\underset{\epsilon \geq 0, A_1, \ldots, A_n}{\text{minimize}} \quad \epsilon + \lambda \sum_{i \in N} \left( A_i - \frac{1}{|N|} A_N \right)^2,$$

$$\text{subject to: } \sum_{i \in N} A_i = A_N,$$

$$\sum_{i \in C_k} A_i \geq A_{C_k}(s, a_{C_k}) - \epsilon, \forall C_k \in \mathcal{C}. \tag{19}$$

The proof of Theorem 4 an approach inspired by Yan & Procaccia (2021), where the core allocation is approximated using sampled coalitions. The key idea is to leverage the properties of the VC-dimension of a function class to bound the probability of deviating from the true allocation in the core. To establish this result, we introduce the following two known lemmas, which play a crucial role in the proof.

Before proving the theorem, we first introduce a lemma regarding the VC-dimension of a function class, as this concept is essential to understanding the behavior of the classifier we employ in the proof.

**Lemma 10.** *Let $\mathcal{F}$ be a function class from $\mathcal{X}$ to $\{-1, 1\}$, and let $\mathcal{G}$ have VC-dimension $d$. Then, with $m = O\left( \frac{d + \log\left(\frac{1}{\Delta}\right)}{\delta^2} \right)$ i.i.d. samples $\{x^1, \ldots, x^m\} \sim \mathcal{P}$, we have:*

$$\left| \Pr_{x \sim \mathcal{P}}[f(x) \neq y(x)] - \frac{1}{m} \sum_{i=1}^{m} \mathbb{1}_{f(x^i) \neq y(x^i)} \right| \leq \delta,$$

*for all $f \in \mathcal{F}$ and with probability $1 - \Delta$.*

This lemma essentially states that if the VC-dimension of a function class is $d$, then by taking a sufficient number of samples $m$, the empirical error rate of a classifier $f$ on those samples is close to the true error rate with high probability (i.e., $1 - \Delta$).

In the context of the theorem, we use linear classifiers to represent the core allocation constraints. The following lemma establishes the VC-dimension of the class of linear classifiers we use.

**Lemma 11.** *The function class $\mathcal{F}^n = \{x \mapsto sign(w \cdot x) : w \in \mathbb{R}^n\}$ has VC-dimension $n$.*

This lemma states that the VC-dimension of the class of linear classifiers is equal to the dimension $n$ of the input space, which is important for bounding the number of samples required to approximate the core allocation effectively.

Now, we combine the insights from the previous lemmas to prove the theorem.

*Proof.* Consider a coalition $C$ sampled from the distribution $\mathcal{P}$. We represent the coalition as a vector $z^C = (z^C, -A_C(s, a_C), 1)$, where $z^C \in \{0, 1\}^n$ is the indicator vector for the coalition and $A_C(s, a_C)$ is the total allocation for the agents not in $C$.

We define a linear classifier $f$ based on parameters $w^f = (A, 1, \epsilon)$, where $w^f \in \mathbb{R}^{n+2}$. The classifier $f(z^C) = \text{sign}(w^f \cdot z^C)$ is designed to capture the core allocation for each coalition $C$.

To ensure coalition rationality, we want the classifier $f$ to satisfy $f(z^C) = 1$ for all coalitions $C \subseteq N$. This ensures that the allocation is in the core for all coalitions. The class of such classifiers is:

$$\mathcal{F} = \{z \mapsto \text{sign}(w \cdot z) : w = (A, 1, \epsilon), A \in \mathbb{R}^n\}.$$

This class of functions $\mathcal{F}$ has VC-dimension at most $n + 2$ by Lemma 11.

Now, solving the quadratic programming problem on $m$ samples of coalitions $\{C_1, \cdots, C_m\}$ provides a solution $(\hat{A}, \hat{\epsilon})$, and the corresponding classifier $\hat{f}$. For each sample coalition $C_k$, we have $\hat{f}(z^{C_k}) = 1$.

By applying Lemma 10, with probability $1 - \Delta$, we obtain the following inequality:

$$\Pr_{C \sim \mathcal{P}} \left[ \sum_{i \in C} \hat{A}_i - A_C(s, a_C) + \hat{\epsilon} \geq 0 \right] \geq 1 - \delta.$$

This shows that the allocation vector generated by solving the quadratic programming problem over the sampled coalitions is within the $\delta$-probable core with high probability (i.e., with probability at least $1 - \Delta$). Thus, Theorem 4 is proved. $\qquad\square$

## A.7 TRUST-REGION DECOMPOSITION AND LOWER BOUNDS ON POLICY IMPROVEMENT

In this section, we discuss based on the definition $A_C(s, a_C) = \mathbb{E}_{a_{N \setminus C} \sim \pi_{N \setminus C}}[Q(s, a_C, a_{N \setminus C})] - V(s)$, and derive Theorem 5 under the centralized TRPO/PPO framework. This section derives theoretical lower bounds on the individual log-probability improvement $\Delta \log \pi_i(a_i \mid s)$ when each agent performs a policy update based on the allocated individual advantage $A_i(s, a)$ for the given sample $(s, a)$.

Under our setting, the joint policy factorizes as $\pi(a \mid s) = \prod_{i \in N} \pi_i(a_i \mid s)$. We aim to maximize the local advantage improvement allocated to each agent while controlling the overall joint KL divergence.

Given action $a = (a_i, a_{-i})$, consider the global policy-update problem:

$$\max_{\{\pi_i'\}} \sum_{i \in N} \mathbb{E}_{a_i \sim \pi_i}[r_i(a_i \mid s) A_i(s, a)] \quad \text{s.t.} \quad \text{KL}(\pi'(\cdot \mid s) \| \pi(\cdot \mid s)) \leq \delta, \tag{20}$$

where $r_i(a_i \mid s) = \pi_i'(a_i \mid s) / \pi_i(a_i \mid s)$.

Due to factorization, the joint KL satisfies

$$\text{KL}(\pi' \| \pi) = \sum_{i \in N} \text{KL}(\pi_i' \| \pi_i),$$

which implies that there is a single global trust-region constraint.

Relaxing the constraint with $\eta > 0$ yields the penalized form:

$$\max_{\{\pi_i'\}} \left( \sum_{i \in N} \mathbb{E}_{a_i \sim \pi_i}[r_i(a_i \mid s) A_i(s, a)] - \frac{1}{\eta} \sum_{i \in N} \text{KL}(\pi_i' \| \pi_i) \right). \tag{21}$$

The objective fully decomposes across $\pi_i'$, producing $n$ independent subproblems:

$$\max_{\pi_i'} \left( \mathbb{E}_{a_i \sim \pi_i}[r_i(a_i \mid s) A_i(s, a)] - \tfrac{1}{\eta} \text{KL}(\pi_i' \| \pi_i) \right), \qquad \forall i. \tag{22}$$

Let $q_i(a_i) = \pi_i'(a_i \mid s)$ and $p_i(a_i) = \pi_i(a_i \mid s)$. The subproblem becomes

$$\max_{q_i} \sum_{a_i} q_i(a_i) A_i(s, a) - \frac{1}{\eta} \sum_{a_i} q_i(a_i) \log \frac{q_i(a_i)}{p_i(a_i)},$$

subject to $\sum_{a_i} q_i(a_i) = 1$.

Construct the Lagrangian

$$\mathcal{L}(q_i, \lambda) = \sum_{a_i} q_i(a_i) A_i(s, a) - \frac{1}{\eta} \sum_{a_i} q_i(a_i) \log(q_i / p_i) + \lambda \left( \sum_{a_i} q_i - 1 \right).$$

Taking the derivative w.r.t. $q_i(a_i)$ and setting it to zero yields $\log(q_i / p_i) = \eta A_i(s, a) + c$. Thus the optimal update is

$$\pi_i'(a_i \mid s) = \frac{\pi_i(a_i \mid s) \exp(\eta A_i(s, a))}{Z_i(s, a_{-i})}, \qquad Z_i = \mathbb{E}_{a_i \sim \pi_i}[\exp(\eta A_i(s, a))], \tag{23}$$

and consequently

$$\Delta \log \pi_i(a_i \mid s) = \eta A_i(s, a) - \log Z_i(s, a_{-i}).$$

If $m_i \leq A_i(s, a) \leq M_i$ (define $R_i = M_i - m_i$), let $X = A_i(s, a)$ with $a_i \sim \pi_i$. Hoeffding's lemma gives

$$\log Z_i \leq \eta \mathbb{E}[X] + \frac{\eta^2 R_i^2}{8}.$$

Hence,

$$\Delta \log \pi_i(a_i \mid s) \geq \eta\big(A_i(s, a) - \mathbb{E}_{a_i \sim \pi_i}[A_i(s, a)]\big) - \frac{\eta^2 R_i^2}{8}. \tag{24}$$

The CORA coalition constraint states that for any $C \subseteq N$,

$$\sum_{j \in C} A_j(s, a) \geq A_C(s, a_C) - \epsilon.$$

It can be proven that $\mathbb{E}_{a_i \sim \pi_i}[A_i(s, a)] \leq \epsilon$. For any given $(s, a)$, we have $A_i(s, a) = A_N(s, a) - \sum_{j \neq i} A_j(s, a)$. This can be rewritten as: $A_i(s, a) \leq A_N(s, a) - A_{N-i}(s, a) + \epsilon$. Since $A_N = Q(s, a) - V(s)$, and $A_C(s, a) = Q_C(s, a) - V(s) = \mathbb{E}_{a_C \sim \pi_C}[Q(s, a)] - V(s)$, we have:

$$\mathbb{E}_{a_i \sim \pi_i}[A_i(s, a)] \leq \mathbb{E}_{a_i \sim \pi_i}[Q(s, a) - V(s)] - \mathbb{E}_{a_i \sim \pi_i}[Q_{N-i}(s, a_{N-i}) - V(s)] + \epsilon.$$

Since $\mathbb{E}_{a_i \sim \pi_i}[Q(s, a)] = Q_{N-i}(s, a_{N-i})$, and $\mathbb{E}_{a_i \sim \pi_i}[Q_{N-i}(s, a)] = Q_{N-i}(s, a_{N-i})$, it follows that:

$$\mathbb{E}_{a_i \sim \pi_i}[A_i(s, a)] \leq 0 + \epsilon.$$

Thus the final lower bound on the individual log-probability improvement becomes

$$\Delta \log \pi_i(a_i \mid s) \geq \eta(A_i(s, a) - \epsilon) - \frac{\eta^2 R_i^2}{8}. \tag{25}$$

The coalition log-probability change is $\Delta \log \pi_C(a_C \mid s) = \sum_{i \in C} \Delta \log \pi_i(a_i \mid s)$. Thus,

$$\Delta \log \pi_C(a_C \mid s) \geq \eta\Big(\sum_{i \in C} A_i(s, a) - |C|\epsilon\Big) - \sum_{i \in C} \frac{\eta^2 R_i^2}{8}. \tag{26}$$

Applying the coalition advantage constraint $\sum_{i \in C} A_i \geq A_C(s, a_C) - \epsilon$ gives a tighter lower bound:

$$\Delta \log \pi_C(a_C \mid s) \geq \eta\big(A_C(s, a_C) - (1 + |C|)\epsilon\big) - \sum_{i \in C} \frac{\eta^2 R_i^2}{8}. \tag{27}$$

Thus, we can derive Theorem 5.

If $A_C(s, a) \geq A_N(s, a) \geq 0$, then $\sum_{i \in C} A_i \geq A_C(s, a) - \epsilon \geq A_N(s, a) - \epsilon$, and

$$\sum_{i \notin C} A_i(s, a) = A_N(s, a) - \sum_{i \in C} A_i(s, a) \leq \epsilon.$$

Thus a high-value coalition $C$ receives almost all advantage $A_N(s, a) - \epsilon$.

If $A_C(s, a) \geq 0 > A_N(s, a)$, then $\sum_{i \in C} A_i \geq A_C(s, a) - \epsilon \geq -\epsilon$, and

$$\sum_{i \notin C} A_i(s, a) = A_N(s, a) - \sum_{i \in C} A_i(s, a) \leq A_N(s, a) + \epsilon.$$

Minimizing $\epsilon$ assigns the coalition $C$ only small cost $-\epsilon$, while $N \setminus C$ absorbs the larger cost $A_N(s, a) + \epsilon$. Thus, even when the global action fails but $A_C(s, a)$ is valuable, the coalition's action probability is preserved—this mechanism is the core driving principle of CORA.

---

**Algorithm 1** Core Advantage Decomposition (CORA)

---

1: **Initialize:** Central critic network $\theta_V$, $\theta_Q$; actor network $\phi_i$ for each agent $i$
2: **for** each training episode $e = 1, \ldots, E$ **do**
3:     Initialize state $s^0$ and experience buffer
4:     **for** each step $t$ **do**
5:         Sample action $a_i^t$ from $\pi_i(a_i|s^t; \phi_i)$ for each agent
6:         Execute the joint action $(a_1^t, \ldots, a_n^t)$
7:         Get reward $r^{t+1}$ and next state $s^{t+1}$
8:         Add data to experience buffer
9:     **end for**
10:    Collate episodes in buffer into a single batch
11:    Compute the target value: $y^t = r(s^t, a^t) + \gamma Q(s^{t+1}; \theta_V)$
12:    **for** $t = 1, \ldots, T$ **do**
13:       Sample $m$ coalitions $\mathcal{C} = \{C_1, \ldots, C_m\} \subseteq 2^N$
14:       **for** each coalition $C \in \mathcal{C}$ **do**
15:          Estimate coalitional advantage $A_C(s^t, a_C^t)$ for each coalition $C \in \mathcal{C}$.
16:          where $\bar{a}_i = \arg\max_{a_i} \pi_i(a_i|s_i^t; \phi_i)$
17:       **end for**
18:       Estimate grand coalition $N$'s advantage $A(s^t, a^t)$ with GAE estimator
19:       Solve the programming problem to obtain credit allocation $\hat{A}_i^t$
20:    **end for**
21:    Update actor networks $\phi_i$ using PPO-clipped policy gradient:

$$\nabla_{\phi_i} \log \pi_{\phi_i}(a_i^t|s_i^t) \cdot \text{clip}\left( \frac{\pi_{\phi_i}(a_i^t|s_i^t)}{\pi_{\phi_i}^{\text{old}}(a_i^t|s_i^t)}, 1 - \epsilon, 1 + \epsilon \right) \cdot \hat{A}_i^t$$

22:    Update critic $\theta_V$ using TD error:   $\sum_t \left( V(s^t; \theta_V) - y^t \right)^2$
23:    Update critic $\theta_Q$ using error:   $\sum_t \left( Q(s^t, a^t; \theta_Q) - y^t \right)^2$
24: **end for**

---

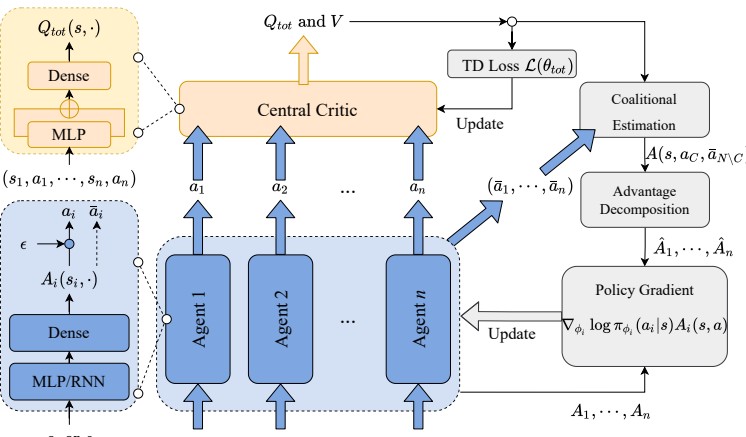

Figure 7: The framework of CORA in Multi-Agent Reinforcement Learning.

## A.8 Algorithm Pseudocode and Diagram

Algorithm 1 outline the implementation of CORA within a standard actor–critic training loop. At each update, a set of coalitions is sampled, and the corresponding coalitional advantages are estimated. A constrained quadratic program is then solved to assign individual credits, which are used to guide policy updates. This procedure ensures that policy gradients reflect coalition-level contributions, encouraging coalitional coordination.

As illustrated in Figure 7, the framework demonstrates the process of Coalitional Advantage Estimation and subsequent Credit Allocation in policy gradient methods.

## A.9 Experimental Details

Table 1: Training Hyperparameters for Each Environment

| Environment | Actor LR | Critic LR | $\gamma$ | GAE $\lambda$ | Clip $\epsilon$ | Parallel Envs |
|---|---|---|---|---|---|---|
| Matrix Games | $5 \times 10^{-4}$ | $5 \times 10^{-3}$ | 0.99 | 0.95 | 0.3 | 4 |
| Differential Games | $5 \times 10^{-5}$ | $5 \times 10^{-4}$ | 0.99 | 0.95 | 0.2 | 4 |
| Multi-Agent MuJoCo | $5 \times 10^{-4}$ | $5 \times 10^{-3}$ | 0.99 | 0.95 | 0.2 | 4 |
| VMAS (Navigation) | $5 \times 10^{-4}$ | $5 \times 10^{-3}$ | 0.99 | 0.95 | 0.2 | 64 |
| VMAS (Others) | $5 \times 10^{-4}$ | $5 \times 10^{-3}$ | 0.99 | 0.95 | 0.2 | 16 |

All experiments were conducted on platforms with AMD 7970X 32-Core CPU, 128GB RAM, and RTX 4090 GPU (24GB). Each algorithm was trained with a two-layer multilayer perceptron (MLP) with a hidden width of 64, except for the *Give-Way* scenario in VMAS, which used a custom network structure. Unless otherwise specified, each configuration was run five times with different random seeds for both the algorithm and the environment. We used the full coalition set ($2^n$ coalitions) for credit assignment across all tasks. For efficiency, 64 parallel environments were used in the *Navigation* task of VMAS, while others used 16 or 4 as listed.

## A.10 Ablation Study of Coalition Sample Size, Std term

To evaluate the impact of coalition sampling size on performance, we conduct an ablation experiment in a differential game environment with 5 agents. Due to the high computational cost in large-scale multi-agent tasks, this experiment focuses on the 5D differential game setting, which balances complexity and tractability.

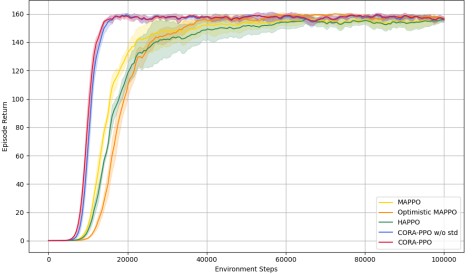 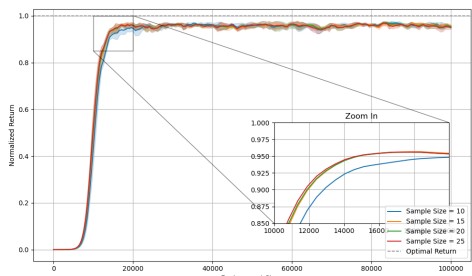

Figure 8: Training performance in the 5D differential game scenario. **Left**: Comparison among baseline methods. **Right**: Effect of coalition sampling size (sample sizes = 10, 15, 20, 25; full coalition size is $2^n - 2 = 30$). All algorithms are repeated 5 times to obtain a 95% confidence interval. Key hyperparameters: Actor learning rate $5 \times 10^{-5}$, Critic learning rate $5 \times 10^{-4}$, $\gamma = 0.99$, GAE $\lambda = 0.95$, 10 epochs per update, clip $\epsilon = 0.2$, and 4 parallel environments.

As shown in Figure 8, increasing the coalition sample size generally improves performance, particularly in the early stages of training, as highlighted in the zoomed-in window. However, even with smaller sampling sizes (e.g., 10 or 15), the CORA algorithm still achieves competitive results. This

indicates that CORA is robust to sample efficiency and remains effective under reduced computation, making it applicable to environments with a moderate number of agents. In addition, CORA with variance often improves performance.

The original credit allocation formulation is a constrained quadratic program, which we relax by linearizing the variance term, resulting in a more efficient linear programming form.

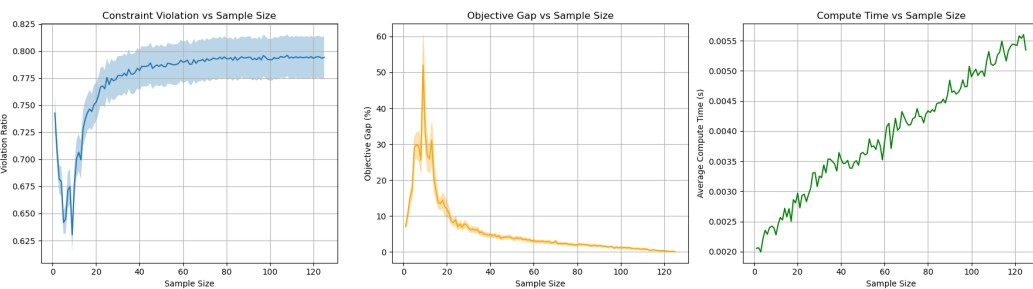

Figure 9: Error and Time Cost of Approximate Credit Assignment. Violation Ratio: proportion of coalition rationality constraints that are violated; Objective Gap: percentage difference in optimization objective compared to the full solution; Compute Time: average runtime across trials. (Number of agents = 7; Advantage functions are randomly generated across 20 trials.)

Figure 9 shows that using only a small number of sampled coalitions yields an accurate and computationally efficient approximation. While constraint satisfaction may degrade slightly with fewer samples, the overall objective gap remains low, and compute time is significantly reduced. This supports the use of approximate credit assignment methods in large-scale scenarios, where full enumeration of $2^n$ coalitions is infeasible.

# B LLM USAGE

We used a large language model (LLM) as an assistive tool for: (i) language editing (grammar and clarity), (ii) consistency checks on LaTeX labels and formatting. The LLM did not generate research ideas, proofs and experimental results. No proprietary or non-anonymized data were provided to the LLM.

