# OpenReview forum: "Core Advantage Decomposition for Policy Gradients in Multi-Agent Reinforcement Learning"
_ICLR.cc/2026/Conference — Submitted to ICLR 2026_

### Official Review · Reviewer_yTEA · 2025-10-29

**Soundness:** 2
**Presentation:** 3
**Contribution:** 2
**Rating:** 2
**Confidence:** 4

**Summary:**

The paper tackles credit assignment in cooperative multi-agent reinforcement learning (MARL). It proposes a core-guided advantage decomposition method grounded in cooperative game theory, where individual agent advantages are derived to satisfy a $\epsilon$-core solution concept. To reduce computational overhead, the method approximates the core via random coalition sampling. The authors provide lower bounds on coalition policy improvement and evaluate on matrix games, VMAS, and cooperative MuJoCo.

**Strengths:**

1. Provides theoretical analysis, including lower bounds on coalition policy improvement.

2. Uses a principled cooperative game-theoretic framework for advantage decomposition.

3. Demonstrates empirical effectiveness on matrix games, VMAS, and cooperative MuJoCo tasks.

**Weaknesses:**

1. Related work and baselines appear outdated.

2. Reported improvements over baselines are modest and not clearly statistically significant.

3. Missing evaluations on widely used benchmarks such as SMAC/SMACv2 and Google Research Football.

4. Assumes additivity (sum of individual advantages equals joint advantage), which may not hold in highly non-linear interactions or with strong coordination requirements.

5. Coalition advantages are estimated for counterfactual coalitions that the critic may not have seen, raising out-of-distribution estimation concerns.

6. Sensitivity to the $\epsilon$-core hyperparameter may affect stability and reproducibility; its selection criteria are unclear.

7. Scalability  of random coalition sampling with increasing agent counts are not fully characterized.

**Questions:**

1. How are counterfactual coalition advantages computed when the critic has not seen those joint actions in the replay buffer? Are there measures to mitigate out-of-distribution bias (e.g., constraints, regularization, uncertainty)?

2. How sensitive is performance to the $\epsilon$ parameter of the $\epsilon$-core?

3. Do you evaluate on SMAC/SMACv2 and Google Research Football? If not, what prevents running on these benchmarks, and how do you expect the method to scale there?

4. How does computational cost scale with number of agents and coalition samples (training/inference wall-clock, GPU hours)? What is the per-update overhead relative to standard CTDE methods?

---

> ### Author Response · Authors · 2025-12-03
>
> Regarding baselines, we focus on MAPPO and HAPPO because they remain widely adopted and competitive CTDE policy-gradient baselines in current MARL literature (including SMAC and MaMuJoCo).
>
> We have already included additional results in both SMAC and Google Football, where CORA-PPO achieved better performance in most map scenarios. In the revision, we report mean performance with 95% confidence intervals across 5–8 random seeds for all environments (see Sec. 6 and Appendix, e.g., matrix games, differential games, VMAS, MaMuJoCo, SMAC, and GRF). In these settings, CORA-PPO not only converges faster but also achieves clearly higher performance than MAPPO/HAPPO.
>
> Regarding the additivity issue, we are not decomposing the value or advantage function; rather, we are designing how to incentivize each agent. We would like to clarify that we are not assuming the environment or the true value function is additive. The additivity constraint is imposed only on the learned credit signals used in the policy gradient update, not on the underlying Q-function or environment dynamics. The critic $Q(s,a)$ and $A_N(s,a)$ can be arbitrarily non-linear in the joint action; CORA does not assume any linear decomposability of $Q$. We enforce $\sum_i A_i = A_N$ for two reasons: (i) to preserve effectiveness in the sense of cooperative game theory and standard value-decomposition methods, and (ii) to keep the policy gradient aligned with the global advantage so that existing TRPO/PPO improvement guarantees still apply (our theoretical analysis in Sec. 4 / Appendix explicitly uses this).
>
> In our paper, we first adopt a constrained form where, when the global advantage is positive, we aim to assign as much advantage as possible to coalitions with potential, while assigning relatively less to other agents, which helps identify better actions. This idea aligns with the core: once we evaluate and allocate advantages across all coalitions, this essentially becomes a core solution.
>
> When extended to broader scenarios, more exploration means that relatively more advantage amplification can be provided to more promising coalitions. Sampling more coalitions typically brings more inference costs, but it also leads to performance improvements. When no coalitions are sampled, regularization ensures that all agents are assigned $\frac{1}{n}A(s, a)$, which is essentially a proportional scaling of $A(s, a)$, similar to MAPPO. In other words, sampling more coalitions can improve performance. In scenarios with a large number of agents, only a limited number of coalitions can be sampled to assess their contributions. The corresponding improvement exists but naturally becomes smaller.
>
> The primary time cost arises during the inference process, rather than from training. We have recorded the process of an experiment. The experiment results are summarized as follows:
> | Scenario/Algorithm | CORA-PPO  | CORA-PPO w/o std | HAPPO     | MAPPO     |
> | ------------------ | --------- | ---------------- | --------- | --------- |
> | Ant2x4             | 1.11 hr   | 1.076 hr         | 1.05 hr   | 58.67 min |
> | Ant4x2             | 1.53 hr   | 1.479 hr         | 1.404 hr  | 59.3 min  |
> | HalfCheetah\_2x3   | 41.77 min | 40.18 min        | 36.8 min  | 34.26 min |
> | Walker2d           | 50.01 min | 47.24 min        | 42.94 min | 40.56 min |
> *(Experiments conducted on a platform with an AMD 7970X 32-Core CPU, 128GB RAM, and RTX 4090 GPU. All possible 2^n coalitions were used for credit assignment.)*
>
> Out-of-distribution estimation concerns: The $A_C$ evaluated by the Critic value is based on the coalition constraints, and we design the advantage values such that $\sum_{i \in C} A_i \geq A_C - \epsilon$, where $\epsilon$ is minimized as much as possible. If the Critic induces an error (e.g., if $A_C$ is too small, then no constraints are formed for $A_i$ where $i \in C$; whereas, if $A_C$ is too large, it will push $A_i$ upwards, leading to an overly large $A_i$, but other coalition constraints also enforce that $A_C - \epsilon \leq \sum_{i \in C} A_i \leq A_N - A_{N \setminus C} + \epsilon$), thus limiting the possible values for the agents. We agree that OOD estimation is a real concern. A more principled treatment (e.g., uncertainty-aware critics (ensemble, bootstrapping) or restricting coalitions to near-on-policy regions) is an important direction for future work.
>
> Sensitivity to the $\epsilon$-core hyperparameter: In our formulation, $\epsilon$ is not a user-tuned hyperparameter.. $\epsilon$ is a slack variable that ensures the assignment problem remains feasible while maximizing the advantage of the coalition agent to reach values near the coalition advantage threshold. In practice, $\epsilon$ decreases progressively during training as the policy matures and $A_C$ diminishes with policy improvement.

---

### Official Review · Reviewer_f3j2 · 2025-10-30

**Soundness:** 3
**Presentation:** 2
**Contribution:** 2
**Rating:** 4
**Confidence:** 3

**Summary:**

The paper introduces CORA, a credit-assignment wrapper for cooperative MARL policy gradients. CORA (i) computes coalition-level advantages for every subset of agents, (ii) allocates individual credits by solving a quadratic program that respects the strong ε-core (coalition rationality + global budget), and (iii) approximates the exponential set of constraints via random coalition sampling. Theoretically, CORA guarantees that beneficial coalitions receive a lower-bound policy improvement even when global advantage is negative. Empirically, CORA consistently outperforms MAPPO, HAPPO and Shapley-based baselines on matrix games, differential games, VMAS and Multi-Agent MuJoCo.

**Strengths:**

1. New granularity: First work to embed core solution (cooperative game theory) inside policy-gradient updates; bridges coalitional stability and MARL credit assignment.
2. Theoretical substance: Novel lower bounds on log-policy improvement for any coalition; shows ε-core ensures provable incentives for valuable sub-teams.
3. Scalable approximation: Random sampling reduces 2ⁿ QP constraints → O(n/δ²) with controllable δ-probable core guarantee (Theorem 4).
4. Strong empirical record: SOTA or near-SOTA on 12 tasks spanning discrete, continuous, dense and sparse-reward settings; ablations verify sample-efficiency and robustness to small coalition budgets.
5. Reproducibility: Complete pseudocode, hyper-parameters, seeds and anonymized code provided; experiments use public benchmarks.

**Weaknesses:**

1. Computational Footprint
   Even with random coalition sampling, CORA requires hundreds of extra Q-value evaluations per step, significantly increasing training cost. The paper does not report wall-clock overhead relative to MAPPO for n = 6, 10, 15, limiting its practical deployment at scale.
2. Scalability Ceiling
   The algorithm’s complexity is O(m·|C| + QP). Current experiments only go up to n = 6 agents, with no results for n ≥ 20, leaving unclear how performance degrades in larger systems.
3. Insufficient Baseline Comparison
   The paper does not compare with recent Shapley-based policy gradient methods (e.g., SHAQ, SCCA) or role decomposition methods like RODE, limiting a full assessment of CORA’s relative strengths.
4. Variance Regularization Issues
   Experiments show that “CORA w/o std” outperforms full CORA in some tasks (e.g., differential games), suggesting that variance regularization may suppress exploration. There is no adaptive mechanism or analysis of when to enable/disable this term.
5. Strong Theoretical Assumptions
   - The provided lower bounds on policy improvement rely on compatible linear critics and small step size α, but no discussion is given for deep neural network critics or non-linear policies.
   - Theorem 4 gives a δ-probable core guarantee, but no rate is provided for how ε decreases with sample size m, lacking insight into the trade-off between approximation quality and sampling efficiency.

**Questions:**

1. Computational Cost & Real-Time Feasibility
   What is the per-step training time and GPU memory usage of CORA compared to MAPPO when n = 10 or 15? Can further parallelization or approximation reduce this cost?
2. Performance at Scale
   How does CORA perform degrade as n ≥ 20? Does the policy improvement lower bound still hold? Are there QP solver failures or insufficient sampling issues?
3. Comparison with Recent Credit Assignment Methods
   How does CORA compare with latest Shapley-based PG methods (e.g., SHAQ, SCCA) or RODE? Under what task structures does CORA offer clear advantages?
4. Adaptive Variance Regularization
   Can an adaptive mechanism be designed to dynamically adjust the strength of the variance regularization term based on task exploration difficulty or training stage?
5. Theory Extension to Deep Critics
   Do CORA’s policy improvement bounds still hold under deep neural network critics and non-linear policies? Can compatible function approximation or other techniques extend the theory?
6. Empirical ε vs. m Relationship
   How does ε decrease as sample size m increases in practice? Can empirical curves or tighter theoretical bounds be provided to guide sampling strategy?

---

> ### Author Response · Authors · 2025-12-03
>
> We sincerely thank the reviewer for the thorough evaluation and constructive suggestions. We have made corresponding theoretical and experimental revisions in the updated manuscript. Our detailed responses to the raised points are provided below.
>
> 1. **Time Cost**: The primary time cost arises during the inference process, rather than from training. When no coalitions are sampled, regularization ensures that all agents are assigned $\frac{1}{n} A(s, a)$, which represents a proportional scaling of $A(s, a)$, similar to the MAPPO approach. Increasing the number of sampled coalitions can enhance performance by improve the coalitional performance. We have recorded the process of an experiment.
>
> | Scenario/Algorithm | CORA-PPO  | CORA-PPO w/o std | HAPPO     | MAPPO     |
> | ------------------ | --------- | ---------------- | --------- | --------- |
> | Ant2x4             | 1.11 hr   | 1.076 hr         | 1.05 hr   | 58.67 min |
> | Ant4x2             | 1.53 hr   | 1.479 hr         | 1.404 hr  | 59.3 min  |
> | HalfCheetah\_2x3   | 41.77 min | 40.18 min        | 36.8 min  | 34.26 min |
> | Walker2d           | 50.01 min | 47.24 min        | 42.94 min | 40.56 min |
>
> *(Experiments conducted on a platform with an AMD 7970X 32-Core CPU, 128GB RAM, and RTX 4090 GPU. All possible 2^n coalitions were used for credit assignment.)* In addition, the ablation results (in Appendix) show that both the empirical $\epsilon$ and the rate of constraint violations decrease steadily as coalition sample increases.
>
> 2. **QP Solver**: The QP solver did not encounter any failures, as we utilized the slack variable $\epsilon$ to ensure feasibility.
>
> 3. **Reproduction of SQDDPG and Shapley Counterfactual Credits Method**: Due to time constraints, we reproduced SQDDPG, Shapley Counterfactual Credits and other credit assignment methods (VDN, QMIX, QTRAN). We compared the results of these algorithms within the SMAC scenario and observed that CORA-PPO outperformed the others in most map scenarios.
>
> 4. **Regularization**: In most cases, regularization is a favorable approach. In the absence of significant constraints, the credit allocation should be more evenly distributed, rather than concentrating on a single agent. In solvers like Gurobi, feasible solutions typically result in the concentration of advantages on a single agent.
>
> 5. **Exploration and Broader Scenarios**: When extended to broader scenarios, increased exploration leads to relatively more amplification of advantage for promising coalitions. In scenarios with a large number of agents, only a limited number of coalitions can be sampled to assess their contributions. While improvements are still observed, they become increasingly marginal as the number of agents grows.
>
> 6. **Theoretical Work**: We have revisited the theoretical aspects of the method. Please refer to Section 5 and Appendix A.7 in the main text for more details. The theoretical derivation presents challenges, particularly due to the coupling of sample data, where the policy gradient for a single sample may differ from the global gradient over all samples. To address this, we have adopted the widely accepted assumption of compatible function approximation. Furthermore, we provide theoretical lower bounds under the PPO/TRPO framework, as detailed in Theorem 5 and discussed in Appendix A.7.
>
> 7. **Argument for the Differences**: In the SQDDPG paper, the authors assume a convex game as a strict condition, ensuring that the Shapley value falls within the core. However, the Shapley value does not always lie within the core. In this paper, we first adopt a constrained approach, allocating as much advantage as possible to promising coalitions when the global advantage value is positive, while assigning relatively less to other agents. This approach facilitates the discovery of better actions. This methodology aligns with the core concept, and once we evaluate and allocate advantages for all coalitions, it directly corresponds to a core allocation.

---

### Official Review · Reviewer_Skdm · 2025-11-01

**Soundness:** 3
**Presentation:** 3
**Contribution:** 2
**Rating:** 4
**Confidence:** 4

**Summary:**

This paper proposes CORA (Core Advantage Decomposition), a novel method for credit assignment in multi-agent reinforcement learning. CORA formulates advantage decomposition as an $\epsilon$-core problem from cooperative game theory, ensuring coalition rationality and balanced credit allocation among agents. Each agent’s advantage $A_i$ is obtained by solving a quadratic program constrained by coalition-level advantages. The method is theoretically justified under natural policy gradient updates and empirically evaluated on matrix games, differential games, VMAS, and MaMuJoCo tasks, showing improved stability and performance over MAPPO and related baselines.

**Strengths:**

- The paper introduces a principled connection between cooperative game theory ($\epsilon$-core) and multi-agent credit assignment, offering a novel theoretical perspective.
- Cross-agent credit assignment is a fundamental problem in multi-agent reinforcement learning, and the paper provides a novel solution to this problem.

**Weaknesses:**

- The method appears computationally expensive, yet the paper does not provide quantitative analysis or discussion on runtime efficiency or scalability.
- The experiments are limited to small and medium-scale environments; more complex benchmarks such as SMAC or Google Research Football are not tested.
- (Minor) Several figures have small fonts and are difficult to read.

**Questions:**

1. Please compare CORA’s coalition-based advantage decomposition with the implicit credit assignment mechanisms in COMA and VDN, as well as with HAPPO’s explicit but globally shared advantage decomposition. How do these different decomposition paradigms differ in terms of stability, scalability, and theoretical grounding?

2. How would you position CORA relative to recent explicit credit assignment approaches such as
   - She et al. (2022) “Agent-Time Attention for Sparse Rewards Multi-Agent Reinforcement Learning,” and
   - Chen et al. (2023) “STAS: Spatial-Temporal Return Decomposition for Multi-Agent Reinforcement Learning”?

3. Given that CORA currently shows promising results mainly in small to medium-scale environments, is there a principled way to improve its computational efficiency so that it can scale to larger agent populations while maintaining its sample-efficiency advantage? If the approach remains limited to few-agent settings, its practical impact might be constrained.

---

> ### Author Response · Authors · 2025-12-03
>
> We thank the reviewer for the detailed feedback and for highlighting both the strengths and the limitations of our work.
>
> 1. COMA focuses on individual counterfactual advantages $A_i(s, a_i)$ with a fixed baseline $a_{-i}$, but does not control how credits behave at the coalition level. VDN (and its variants) decompose Q-values monotonically for decentralized execution. HAPPO stabilizes learning via sequential per-agent trust-region updates, yet all agents still share the same global advantage $A_N(s, a)$. In contrast, CORA explicitly enforces coalition-level constraints $\sum_{i \in C} A_i(s,a) ;\ge; A_C(s,a_C) - \epsilon$, ensuring that high-value coalitions are systematically favoured in the policy gradient, even when the global joint advantage (A_N(s,a)) is weak or slightly negative. Intuitively speaking, when faced with a higher global advantage value and the potential advantage of an alliance is high, a higher advantage value is assigned to it, but it will not exceed the global advantage value.
>
> 2. The time cost mainly arises during the inference process. Sampling more coalitions typically increases the inference cost, but it also brings performance improvements. When no coalitions are sampled, regularization ensures that all agents are assigned an equal share of the advantage, i.e., $\frac{1}{n}A(s, a)$, which is essentially a proportional scaling of $A(s, a)$, similar to MAPPO. In other words, sampling more coalitions can enhance the performance but it increases the time cost. To quantify overhead, we include real wall-clock measurements below (same codebase, same hardware):
>
> | Scenario/Algorithm | CORA-PPO  | CORA-PPO w/o std | HAPPO     | MAPPO     |
> | ------------------ | --------- | ---------------- | --------- | --------- |
> | Ant2x4             | 1.11 hr   | 1.076 hr         | 1.05 hr   | 58.67 min |
> | Ant4x2             | 1.53 hr   | 1.479 hr         | 1.404 hr  | 59.3 min  |
> | HalfCheetah\_2x3   | 41.77 min | 40.18 min        | 36.8 min  | 34.26 min |
> | Walker2d           | 50.01 min | 47.24 min        | 42.94 min | 40.56 min |
>
> *(Experiments conducted on a platform with an AMD 7970X 32-Core CPU, 128GB RAM, and RTX 4090 GPU. All possible 2^n coalitions were used for credit assignment.)*
> CORA introduces a moderate overhead but remains within the same practical range as HAPPO.
>
> 3. We have already included additional experimental results on SMAC, and also added results on Google Football. CORA-PPO shows better performance across most map scenarios. In a limited time, we reproduced SQDDPG, Shapley Counterfactual Credits and other credit assignment methods (VDN, QMIX, QTRAN). We compared the results of these algorithms on the SMAC scenario, and CORA-PPO performed better in most map scenarios.
>
> 4. There are indeed some challenges in the theoretical proof. Due to the coupling of sample data, there is a discrepancy between the policy gradient on a single sample and the global gradient across all samples. Therefore, we adopted the widely used assumption of compatible function approximation. We have made further progress by providing theoretical lower bounds under the PPO/TRPO framework (see Theorem 5 and the discussion in Appendix A.7). We mainly compared CORA with credit-assignment methods and highlighted the differences: In the SQDDPG paper, the authors assume a convex game as a strict condition, which ensures that the Shapley value falls within the core. However, the Shapley value does not always fall within the core.
>
> 5. In the beginning of this paper, we first adopt a constrained approach. When the global advantage value is positive, we try to allocate as much advantage as possible to promising coalitions, while allocating relatively less to other agents, helping to discover better actions. This approach aligns with the concept of the core, and once we evaluate all the coalitions and allocate accordingly, it turns out to be exactly the core allocation.
>
> 6. Literature: CORA focuses on solving the "coalition-level synergy/credit assignment" problem, where agents must collaborate in specific subsets to succeed. Unlike ATA, which tackles sparse or delayed rewards by redistributing global rewards to individual agents using an Agent-Time attention model, and STAS, which decomposes global returns temporally and spatially (agent-level) using Shapley values, CORA operates at the coalition level. It is better suited for tasks involving strong cooperation among agents rather than simple additive or independent behaviors. CORA uses an explicit constraint framework to allocate credit based on coalition contributions, offering a more solid theoretical foundation in environments requiring intricate coordination. Both ATA and STAS have been cited in our work for comparison.

---

### Official Review · Reviewer_g7Pg · 2025-11-02

**Soundness:** 2
**Presentation:** 2
**Contribution:** 2
**Rating:** 2
**Confidence:** 3

**Summary:**

The paper proposes Core Advantage Decomposition (CORA), a credit-assignment scheme for cooperative MARL that decomposes the global advantage into per-agent credits by solving, at each update, a strong
$\epsilon$-core optimization over coalitions. Concretely, the authors define a coalitional advantage and allocate individual advantages via a quadratic program that enforces core constraints and penalizes deviation from uniform sharing. They also give a sampling-based approximation with a PAC-style guarantee using VC-dimension arguments. CORA is integrated into an actor-critic (PPO-style) training loop with two critics, and is evaluated on matrix games, differential games, VMAS, and MA-MuJoCo. Empirically, CORA reportedly improves returns over MAPPO/HAPPO/COMA in several tasks.

**Strengths:**

1. Casting per-agent advantage allocation as a strong $\epsilon$-core program (Eq. 7) is neat and leads to interpretable coalition rationality constraints and a variance-regularized objective.

2. Theorems 2 and 3 relate NPG updates to (coalitional) improvement, clarifying when beneficial coalitions are amplified even if the global advantage is negative.

3. Theorem 4 provides a simple sample-complexity bound for entering a 𝛿-probable core using VC-dimension, which is rarely discussed in MARL credit assignment.

4. Experiments span matrix/differential games, VMAS, and MA-MuJoCo, with ablations on coalition sampling and discussions of runtime/constraint violations in a synthetic setting.

**Weaknesses:**

1. There is extensive prior work using cooperative-game concepts for credit (e.g., Shapley-based SQDDPG, Shapley Counterfactual Credits, SHAQ). The paper does not compare against these or thoroughly argue why the core yields better learning dynamics than Shapley-style alternatives. The theoretical pieces largely restate core feasibility rather than showing tighter improvement bounds or variance reductions over Shapley. Empirically, none of these Shapley baselines are included.

2. The lower-bound results assume natural policy gradient updates with compatible function approximation, while the implementation uses PPO with clipping and two critics. The paper asserts first-order relations, but does not quantify how clipping, advantage normalization, or off-policy bootstrap in Q affect the guarantees. Rhis leaves the theoretical relevance to the practical algorithm unclear.

3. While MAPPO/HAPPO/COMA are included, other strong contemporary PPO-style baselines and trust-region variants on MA-MuJoCo (e.g., HAPPO/HATRPO references) suggest nuanced performance differences. The paper does not situate its results in that evolving landscape or evaluate on common benchmarks like SMAC or MPE where credit-assignment is heavily studied.

4. The baseline action is chosen as the modal/mean policy output. This may bias estimates and reduce exploration, yet the paper reports that removing the std term helps on differential games, suggesting sensitivity that is not analyzed. Moreover, solving Eq. 7 requires many Q evaluations per step. The cost is discussed only in a random-advantage toy experiment, not the real tasks.

**Questions:**

1. Can you include direct comparisons to SQDDPG, Shapley Counterfactual Credits, and/or SHAQ on at least one VMAS and one MA-MuJoCo task? If not, please justify why these are out of scope and provide a discussion on when core vs.

2. For VMAS and MA-MuJoCo, what is the average per-update time and memory overhead vs. MAPPO/HAPPO when (i) using all coalitions and (ii) using sampled coalitions at your recommended
$m$? Please also report the number of 𝑄 forward passes per batch and the resulting throughput.

3. Have you tried expectation baselines $Q(s,a_C,\pi_{N\backslash C})$ or action-masking variants (as mentioned in Remark 1)? A targeted ablation isolating the effect of the baseline actioncould clarify whether the gains stem from the core constraints or the baseline choice.

---

> ### Author Response · Authors · 2025-12-03
>
> Thank you for the careful reading and valuable feedback. We have revised the manuscript PDF. Please find our specific responses below.
>
> 1.In response to other reviewers' requests for SMAC environment, we compared the performance of these algorithms on the SMAC benchmark, where CORA-PPO outperformed these methods (SQDDPG, Shapley Counterfactual and other credits assignment methods) in most of the map scenarios. Regarding the differences between these methods: In the SQDDPG paper, the authors assume the game is convex, which is a stringent condition, ensuring that the Shapley value falls within the core. In general MARL environments—especially ones with non-monotone interaction like differential games—these assumptions do not hold. CORA directly enforces per-sample coalition constraints $ \sum_{i\in C}A_i(s,a) \ge A_C(s,a_C) - \epsilon$, providing guarantees Shapley does not inherently provide. Shapley values are ex-ante fair, while CORA is per-sample policy-gradient aligned, meaning that (i) it preserves ascent direction and (ii) reinforces high-value coalitions at action level.
> In the beginning of this idea, we first adopt a constrained approach. When the global advantage value is positive, we try to allocate as much advantage as possible to promising coalitions, while allocating relatively less to other agents, helping to discover better actions. This approach aligns with the concept of the core, and once we evaluate all the coalitions and allocate accordingly, it turns out to be exactly the core allocation.
>
> 2.There are indeed some challenges in the theoretical derivation. Due to the coupling of sample data, the policy gradient on a single sample may differ from the global gradient across all samples. To address this, we adopted the widely accepted assumption of compatible function approximation. We initially used a modal/mean action for computational efficiency. In the revised version we discussed the expected-value baseline $A_C(s,a_C) = \mathbb{E}*{a*{N\setminus C}\sim\pi_{N\setminus C}} [Q(s,a_C,a_{N\setminus C})] - V(s)$,  which removes action-level bias entirely. Additionally, we have taken a further step by providing theoretical lower bounds under the PPO/TRPO framework (see Theorem 5 and the discussion in Appendix A.7).
>
> 3.We have already evaluated our method on common benchmarks like SMAC, and we have also conducted additional experiments on Google Research Football. The results indicate that CORA-PPO performs well across most map scenarios, demonstrating its effectiveness in the credit-assignment context.
>
> 4.Baseline choice:  In the revised version we discussed the expected-value baseline $A_C(s,a_C) = \mathbb{E}*{a*{N\setminus C}\sim\pi_{N\setminus C}} [Q(s,a_C,a_{N\setminus C})] - V(s)$,  which removes action-level bias entirely. However, in continuous task experiments, the method is not as good, so we mainly adopt a specific baseline ($\bar{a}_i$ with max prob) in the experiment. For discrete actions, we usually take the Monte Carlo sampling number K=32, which undoubtedly increases the time overhead. Removing the std term: Differential games have highly multimodal joint reward surfaces. The variance penalty over-regularizes and suppresses exploration in these tasks. In addition, the variance term is primarily used for regularization. Without the variance regularizer, the linear constraints admit many degenerate core allocations, and standard QP/LP solvers often return extreme points that concentrate almost all credit on a single agent. This is undesirable for learning stability, which is why we add the variance term to encourage more balanced solutions.
>
> 5.Time Cost: The primary time cost arises during the inference process, rather than from training. Sampling more coalitions typically incurs higher inference costs, but it is accompanied by performance improvements. When no coalitions are sampled, regularization ensures that all agents are assigned $\frac{1}{n} A(s, a)$, which represents a proportional scaling of $A(s, a)$, similar to the MAPPO approach. We have recorded the process of an experiment. The experiment results are summarized as follows:
> | Scenario/Algorithm | CORA-PPO  | CORA-PPO w/o std | HAPPO     | MAPPO     |
> | ------------------ | --------- | ---------------- | --------- | --------- |
> | Ant2x4             | 1.11 hr   | 1.076 hr         | 1.05 hr   | 58.67 min |
> | Ant4x2             | 1.53 hr   | 1.479 hr         | 1.404 hr  | 59.3 min  |
> | HalfCheetah\_2x3   | 41.77 min | 40.18 min        | 36.8 min  | 34.26 min |
> | Walker2d           | 50.01 min | 47.24 min        | 42.94 min | 40.56 min |
> *(Experiments conducted on a platform with an AMD 7970X 32-Core CPU, 128GB RAM, and RTX 4090 GPU. All possible 2^n coalitions were used for credit assignment.)*

---

### Meta-Review · Area_Chair_rSzu · 2025-12-24

**Summary:**

The paper proposes CORA, a method for credit assignment in Multi-Agent Reinforcement Learning (MARL) that uses the "Core" concept from cooperative game theory to decompose global advantage into individual credits. The authors formulate this as a quadratic programming (QP) problem with coalition rationality constraints and employ random sampling for approximation.
While the reviewers acknowledged the novelty of integrating the Core solution into policy gradients and the theoretical attempt to bound coalitional improvement, the consensus leans towards rejection. The primary concerns informing this decision are:

1.	The method requires solving a QP optimization at each step. Even with random sampling, the inference time overhead is non-trivial compared to baselines. Reviewers remained skeptical about the method's feasibility in environments with a large number of agents.

2.	Initial reviews highlighted a lack of comparisons to relevant Shapley-based credit assignment methods and standard benchmarks like SMAC. While the authors added some of these during the rebuttal, the depth of comparison and the "rushed" nature of these additions did not fully alleviate concerns regarding whether CORA offers a significant advantage over existing SOTA methods given its added complexity.

3.	There is a disconnect between the theoretical analysis and the practical implementation. Furthermore, the reliance on estimating counterfactual coalitional advantages raises valid concerns about out-of-distribution (OOD) bias which were not fully resolved.

**Reviewer Concerns:**

Addressed by Rebuttal:

•	The authors addressed the request for standard benchmarks by adding experiments on SMAC and Google Research Football (GRF).

•	The authors provided a table detailing wall-clock training times, acknowledging that CORA introduces overhead.

•	The authors clarified that slack variables are used to prevent solver failures.

Outstanding / Partially Addressed:

•	While SMAC results were added, Reviewers noted the lack of deep comparison with recent Shapley-based methods in the initial submission. The authors "reproduced" these in a limited time, but the thoroughness of this comparison and the specific advantages of the Core over Shapley values in dynamic learning contexts remain debatable.

•	Reviewers raised concerns about scaling. The scalability of the QP approach remains a significant bottleneck.

•	Reviewer raised a critical point about estimating advantages for counterfactual coalitions not seen in the replay buffer. The authors admitted this is a "real concern" but relegated it to future work.

**Reviewer Scores:**

Reviewer g7Pg: Remains. The reviewer requested comparisons to Shapley baselines and SMAC. While the authors provided these, the reviewer's fundamental concern about the theoretical relevance and the high cost of QP solving likely prevents a positive score.

•  Reviewer Skdm: Remains. This reviewer was on the fence. The addition of SMAC/GRF addresses the benchmark issue, but the provided runtime data confirms the method is slower, and the scalability to large N remains unsolved.

•  Reviewer f3j2: Remains. The rebuttal confirmed the computational overhead and did not provide results for large N, validating the reviewer's hesitation regarding practical deployment.

•  Reviewer YTEA: Remains. This reviewer had strong concerns about OOD bias and the additivity assumption. The authors acknowledged the OOD issue is valid but offered no immediate solution, which likely keeps the score below the acceptance threshold.

---

### Decision · Program_Chairs · 2026-01-26

Reject